# Physical functioning in the lumbar spinal surgery population: A systematic review and narrative synthesis of outcome measures and measurement properties of the physical measures

Katie L. Kowalski[1,2]*, Jai Mistry[1,3], Anthony Beilin[1], Maren Goodman[4], Michael J. Lukacs[1,5], Alison Rushton[1]

1 School of Physical Therapy, Western University, London, Ontario, Canada, 2 Collaborative Specialization in Musculoskeletal Health Research, Bone and Joint Institute, Western University, London, Ontario, Canada, 3 Physiotherapy, St George's Hospital, London, United Kingdom, 4 Western Libraries, Western University, London, Ontario, Canada, 5 Physiotherapy Department, London Health Sciences Centre, London, Ontario, Canada

* kkowals7@uwo.ca

**Data Availability Statement:** All relevant data are within the manuscript and its Supporting

## Abstract

### Background

International agreement supports physical functioning as a key domain to measure interventions effectiveness for low back pain. Patient reported outcome measures (PROMs) are commonly used in the lumbar spinal surgery population but physical functioning is multidimensional and necessitates evaluation also with physical measures.

### Objective

1) To identify outcome measures (PROMs and physical) used to evaluate physical functioning in the lumbar spinal surgery population. 2) To assess measurement properties and describe the feasibility and interpretability of physical measures of physical functioning in this population.

### Study design

Two-staged systematic review and narrative synthesis.

### Methods

This systematic review was conducted according to a registered and published protocol. Two stages of searching were conducted in MEDLINE, EMBASE, Health & Psychosocial Instruments, CINAHL, Web of Science, PEDro and ProQuest Dissertations & Theses. Stage one included studies to identify physical functioning outcome measures (PROMs and physical) in the lumbar spinal surgery population. Stage two (inception to 10 July 2023) included studies assessing measurement properties of stage one physical measures. Two

information files. No identifying information is
found within the manuscript or Supporting
information files.

**Funding:** The author(s) received no specific
funding for this work.

**Competing interests:** The authors have declared
that no competing interests exist.

independent reviewers determined study eligibility, extracted data and assessed risk of bias
(RoB) according to COSMIN guidelines. Measurement properties were rated according to
COSMIN criteria. Level of evidence was determined using a modified GRADE approach.

### Results

Stage one included 1,101 reports using PROMs (n = 70 established in literature, n = 67
developed by study authors) and physical measures (n = 134). Stage two included 43 arti-
cles assessing measurement properties of 34 physical measures. Moderate-level evidence
supported sufficient responsiveness of 1-minute stair climb and 50-foot walk tests, insuffi-
cient responsiveness of 5-minute walk and sufficient reliability of distance walked during the
6-minute walk. Very low/low-level evidence limits further understanding.

### Conclusions

Many physical measures of physical functioning are used in lumbar spinal surgery popula-
tions. Few have investigations of measurement properties. Strongest evidence supports
responsiveness of 1-minute stair climb and 50-foot walk tests and reliability of distance
walked during the 6-minute walk. Further recommendations cannot be made because of
very low/low-level evidence. Results highlight promise for a range of measures, but pro-
spective, low RoB studies are required.

### Introduction

Musculoskeletal low back pain (LBP) persists as a leading global cause of disability from ado-
lescence to old age [1]. It is the most prevalent condition requiring effective rehabilitation [2]
with best-evidence guidelines recommending interventions focused on self-management,
physical and psychological therapies [3, 4]. For appropriate clinical indications, surgical inter-
ventions are effective in reducing pain and enhancing physical functioning [3, 5]. Selecting
appropriate outcome measures for the lumbar spinal surgery population is important as popu-
lation-specific outcome measures are recommended for use when measuring treatment out-
comes for specific clinical populations and when focusing on the individual, an important
component of providing patient-centered care [6, 7].

International agreement supports physical functioning as the most important outcome
domain to measure effectiveness of interventions for LBP [8]. Physical functioning is the
impact of a condition on physical activities of daily living (e.g., walking ability, performance
status, disability index) [9]. Use of patient reported outcome measures (PROMs) to evaluate
physical functioning in LBP is common, despite low to very low quality evidence for their con-
tent validity [10]. The Oswestry Disability Index (ODI) is most commonly used/recommended
in LBP and lumbar spinal surgery [11, 12]. However, previous systematic reviews have
highlighted a breadth of PROMs are used to evaluate physical functioning in LBP and lumbar
spinal surgery populations [12, 13]. As these systematic reviews were either conducted over 20
years ago [13] or with a limited search strategy [12], a contemporary and comprehensive
search of the literature will enable wider considerations of PROMs to evaluate physical func-
tioning in lumbar spinal surgery populations.

Physical functioning is a multidimensional construct and necessitates evaluation with
physical outcome measures, including impairments (e.g., strength), performance on a

standardized task (e.g., 6-minute walk) and activity in a natural environment (e.g., step count) [14, 15]. Physical measures (e.g., time to symptom onset during 6-minute walk) are the measurement unit of interest. Physical measures are also an important component of assessment in LBP, informing clinical reasoning to formulate a diagnosis, prognosis and intervention plan. Despite growing widespread use in lumbar spinal surgery [16] and recognized value in other musculoskeletal conditions [17–19], recommendations for physical measures of physical functioning in the lumbar spinal surgery population do not exist.

Selecting outcome measures with adequate measurement properties is key for accurate interpretation of information gained during a clinical assessment and for measuring effectiveness of interventions in research and clinical practice. The COnsensus-based Standards for the selection of health Measurement Instruments (COSMIN) initiative aims to facilitate selecting high-quality outcome measures through a systematic evaluation of validity, responsiveness, reliability and measurement error of PROMs and physical measures [20, 21]. Use of outcome measures with adequate measurement properties supports clinicians in their clinical reasoning for accuracy in assessment and diagnosis, monitoring patient progress, evaluating treatment outcomes and making informed decisions to optimize patient outcomes. When selecting outcome measures, COSMIN also recommends considering interpretability and feasibility, as these are important for clinical understanding of outcome measure scores and application within the local clinical or research context [20]. While use of physical measures in lumbar spinal surgery has risen exponentially [16], there is no systematic review evaluating measurement properties of physical measures of physical functioning in the lumbar spinal surgery population. While systematic reviews of PROM measurement properties exist [12, 22], there is also no contemporary comprehensive resource outlining all PROMs of physical functioning, beyond the ODI, in this population.

## Objectives

1. To identify outcome measures (patient reported and physical) used to evaluate physical functioning in the lumbar spinal surgery population.

2. To assess the measurement properties and describe the interpretability and feasibility of physical measures of physical functioning in the lumbar spinal surgery population.

## Methods

### Design

Using a two-staged approach, this systematic review was conducted according to a registered (PROSPERO CRD42021293880) and published protocol [23]. Stage one identified PROMs (excluding ODI) and physical measures used to evaluate physical functioning in the lumbar spinal surgery population. Results informed the stage two search strategy to identify studies of measurement properties of the physical measures, guided by COSMIN methodology [20, 21]. Reporting aligns with the Preferred Reporting Items for Systematic Review and Meta-Analysis (PRISMA) statement [24]. Ethical approval was not required for this systematic review.

### Eligibility criteria

**Population.** Adults aged ≥18 years listed for or previous lumbar spinal surgery for musculoskeletal LBP and/or low back-related leg pain.

**Intervention.** Lumbar spinal surgery at one or more levels, including thoracolumbar and lumbosacral. Surgery due to trauma, fracture, space occupying mass (e.g., tumor), inflammatory conditions, infection, osteoporosis, congenital scoliosis, cauda equina syndrome and extra-spinal causes of back and/or leg pain were excluded.

**Comparator.** Not applicable.

**Outcome measures.** Stage one: Outcome measures evaluating physical functioning categorized as:

1. PROMs: questionnaires, scales or subscales assessing ≥1 aspects of physical functioning. ODI was excluded as it is a well-established PROM of physical functioning in lumbar spinal surgery [11, 25].

2. Impairment-based: structure or function of a specific body part or system [15] (e.g., strength)

3. Performance-based: performance on a standardized task [14, 26] (e.g., 6-minute walk)

4. Activity in a natural environment: remote monitoring of physical functioning in a natural environment [14, 27] (e.g., step count)

Stage two: Physical outcome measures (categorizations 2–4). Outcome measures not practical within Physical Therapy settings (clinical, hospital, community) were excluded (e.g., imaging).

**Study design of included studies.** Stage one: All study designs and article types.

Stage two: Studies of measurement properties (validity, responsiveness, reliability, measurement error). Studies were excluded if data were not original (e.g., systematic review), normative only or insufficient (e.g., conference abstract).

For both stages, studies not in English were excluded.

## Information sources

Searches were developed in MEDLINE (Ovid) and a librarian (MG) adapted for EMBASE (Ovid), Health and Psychosocial Instruments (Ovid), CINAHL (EBSCOhost), Web of Science Core Collection, Scopus, PEDro (stage one only), and ProQuest Dissertations and Theses. Electronic databases were searched from inception to December 15, 2021 for stage one and inception to July 10, 2023 for stage two. Reference lists of included studies in stage two were hand-searched independently by two authors (KK, JM) to identify additional potential articles.

## Search strategy

Search strategies were developed in collaboration with a librarian (MG; S1 Appendix) and informed by National Institute for Health and Care Excellence (NICE) guidelines for LBP and sciatica over 16s [28]. Stage two also included physical outcome measures identified in stage one and the COSMIN sensitive search and exclusion filter [29]. An independent librarian peer-reviewed stage one search using the Peer Review of Electronic Search Strategies (PRESS) checklist.[30]

## Selection process

Citations were imported into Covidence (Veritas Health Innovation, Australia) and duplicates removed. Title/abstract screening was performed independently in duplicate (KK, JM, AB). Full texts were obtained and reviewed independently in duplicate (KK, JM, AB) for articles meeting eligibility criteria or when eligibility was unclear. Disagreements at each stage were discussed, and a third author (AR) used if consensus was not achieved.

## Data collection process and data items

Data were extracted independently in duplicate (KK, JM, AB) using standardized data extraction sheets. Data extraction included study characteristics, participant characteristics and outcome measures. Stage two was guided by the COSMIN scoring form, which also included measurement properties and information related to interpretability and feasibility, as recommended by COSMIN to aid selection of physical measures [20]. Differences in data extraction were resolved through discussion. One investigator [31] was contacted and responded to one email to clarify reporting during stage two, in accordance with the a priori strategy for contacting study authors [23].

## Risk of bias (RoB) in individual studies

As planned [23], RoB was not assessed in stage one. For stage two, RoB was assessed independently in duplicate (KK, JM) using the COSMIN Checklist [32] and extended tool for measurement instruments [21]. For each study of a measurement property, RoB was rated as "Very good", "Adequate", "Doubtful", or "Inadequate" and the overall rating was determined using the worst score counts principle [21, 32]. Studies using hypothesis testing approaches that did not define a hypothesis or no hypothesis could be derived were rated as "Inadequate" because of strong potential for selective reporting of analyses and outcomes [33, 34]. Disagreements were discussed, and if consensus not achieved, a third author (AR) was used.

## Data synthesis

For stage one, PROMs and physical measures were categorized according to established frameworks of physical functioning. PROMs were categorized according to the IMMPACT/OMERACT (Initiative on Methods, Measurement, and Pain Assessment in Clinical Trials, Outcome Measures in Rheumatoid Arthritis Clinical Trials) framework: general, site-specific, disease-specific, pain-related physical functioning/activities or activities of daily living [14].Physical measures were categorized according to level-two categories of the International Classification of Functioning, Disability and Health (ICF) [15].

For stage two, results of each measurement property study were rated as sufficient (+), insufficient (-) or indeterminant (?) criteria for good measurement properties (S2 Appendix) [20]. Studies using a criterion approach were considered criterion validity or responsiveness [32]. Studies using hypothesis testing approaches that did not define a hypothesis or no hypothesis could be derived were rated as indeterminant. Standards for assessing a priori hypotheses in hypothesis testing approaches have been removed from updated COSMIN guidelines with recommendations the systematic review team formulate hypotheses to evaluate results [20, 32]. However, as most studies did not define a hypothesis or no hypothesis could be derived, formulating post-hoc hypothesis for authors of included studies would have elevated RoB in this systematic review to an unacceptable level as lack of a priori hypotheses introduces threats to the internal validity of included studies and therefore this systematic review would have provided an inaccurate representation of the quality of the literature [35, 36].

High heterogeneity and RoB directed a qualitative synthesis, in accordance with the a priori protocol [23]. Summarized results were rated as sufficient or insufficient if at least 75% of individual studies were rated as sufficient or insufficient, indeterminant if at least 75% of individual studies were rated as indeterminant or inconsistent (±) if less than 75% of the individual studies agreed. Information related to interpretability and feasibility are described, in accordance with COSMIN recommendations [20].

### Reporting biases

Assessment of reporting bias was conducted through evaluating consistency between published results and study protocols, if identified in stage two.

### Overall quality of evidence

Quality of evidence was evaluated for each measurement property per physical measure, using GRADE (Grading of Recommendations Assessment, Development and Evaluation) modified for measurement properties [20]. Four factors contributed to determining quality of evidence (RoB, inconsistency, imprecision, indirectness). Two reviewers (KK, JM) independently determined quality of evidence and disagreements were resolved through discussion. If consensus was not achieved, a third author (AR) was used.

## Results

The PRISMA flow diagram (Fig 1) shows both stages of searching, selection and reasons for exclusion (S3 Appendix). For stage one, complete agreement was achieved between reviewers. For stage two, there was strong agreement between reviewers for title/abstract screening (κ = 0.85) and full text review (κ = 0.94) [37]. Complete agreement on eligibility was achieved through discussion. Due to unclear reporting, the third reviewer (AR) was consulted once about one study [38] to agree which measurement property was investigated.

### Stage one: Identify physical functioning outcome measures

**Study characteristics.** Stage one included 1,101 reports, published across 47 countries over 40 years (1982–2022) with increasing annual publications (S4 Appendix). The age of lumbar spinal surgery populations rose from early 40s in the 1980s to early 60s in the 2010s. For studies that reported sex/gender, reporting was binary and there was about equal representation of male/men and female/women. Common surgical procedures included fusion, decompression and discectomy. Physical functioning was assessed using PROMs in n = 964 reports and physical measures comprising impairments (n = 92 reports), performance (n = 198 reports) and activity in a natural environment (n = 42 reports; S4 Appendix). Most reports included >1 measure of physical functioning.

**Results of synthesis.** *PROMs.*

70 established PROMs were identified and authors from 49 articles developed 67 of their own PROMs in physical functioning categories [14] of (S5 Appendix):

- General (Established: n = 31, 44%; Developed: n = 41, 61%), including subcategories: physical activity, health status/quality of life, functional status, disability, patient-identified functional limitations

- Site-specific (Established: n = 26, 37%; Developed: n = 3, 4.5%), including subcategories: low back and/or leg pain, back/headache/facial pain

- Disease-specific (Established: n = 9, 13%; Developed: n = 1, 1.5%), including subcategories: spinal stenosis, scoliosis

- Pain-related (Established: n = 2, 3%; Developed: n = 2, 3%), including subcategory: pain-related disability not linked to a body region or condition/disease

- Activities of daily living (Established: n = 2, 3%; Developed: n = 20, 30%), including subcategories: self-care. activities of daily living

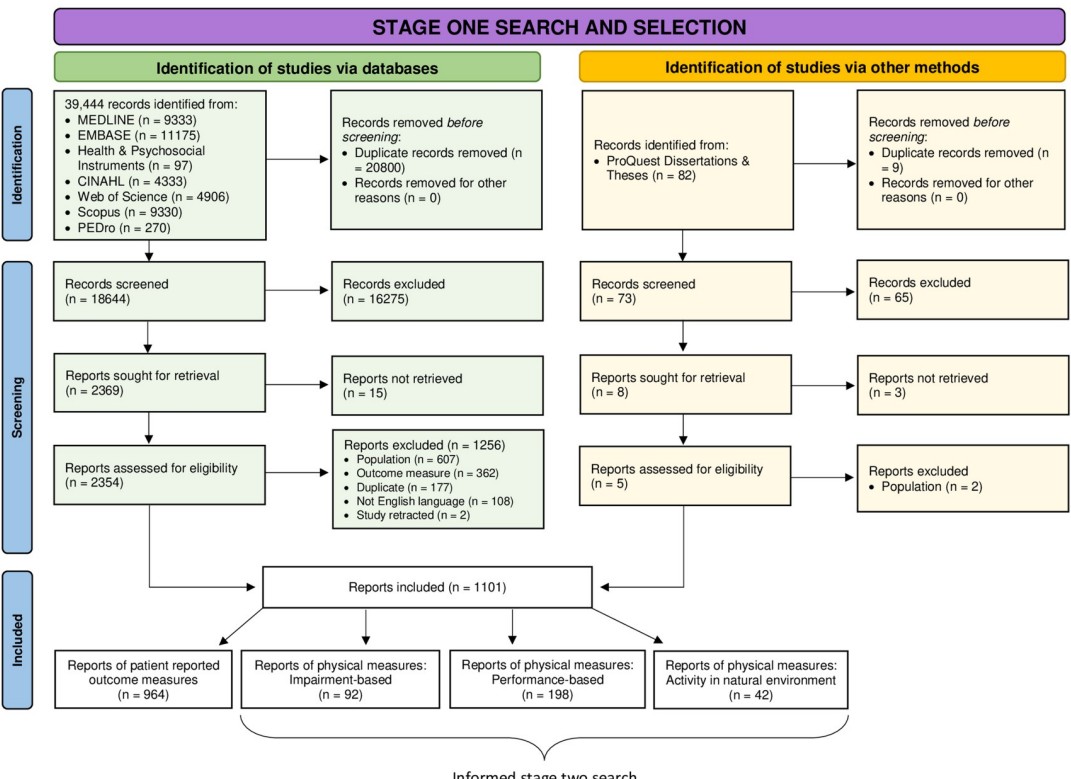

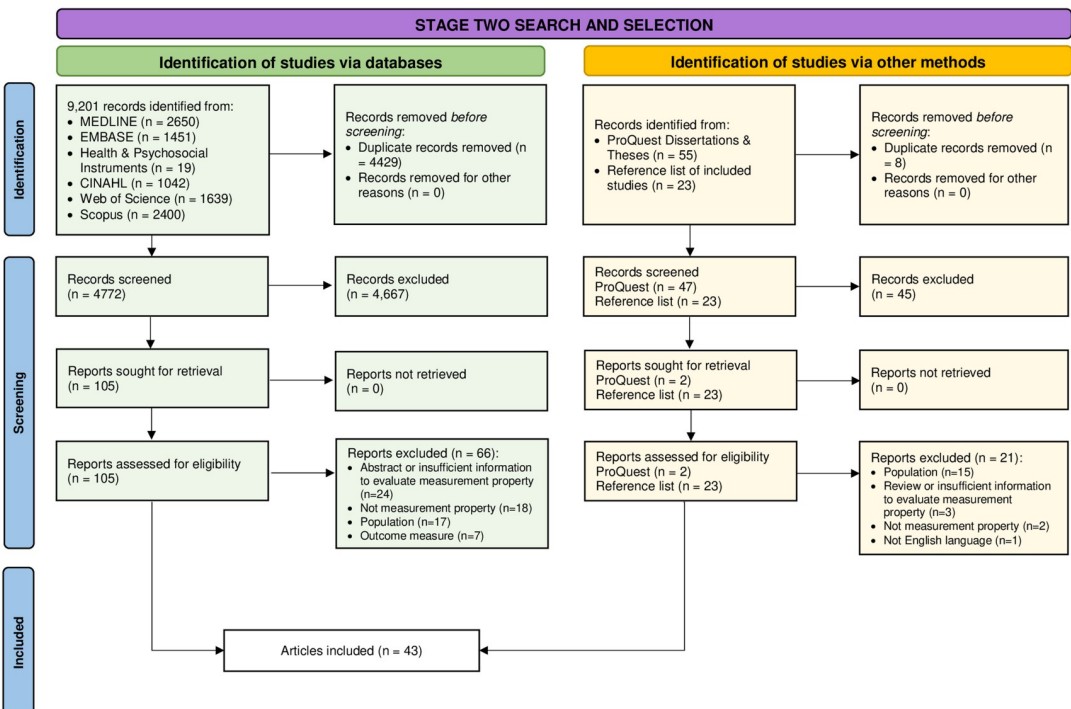

**Fig 1. PRISMA flow diagram of stage one and two search and selection processes.**

The most frequently used PROM to evaluate physical functioning was the Short Form Health Survey (e.g., SF-36, SF-12) physical component score and physical functioning domain (S4 Appendix). The Roland Morris Disability Questionnaire was the second most frequently used PROM with several versions and modifications (24-item, 23-item, for sciatica, substitute 'leg pain' for 'back pain', modifications not further specified). The importance of walking in the lumbar spinal surgery population is highlighted by 23 different stand-alone question/response option combinations to measure walking capacity.

### Physical measures

134 physical measures were identified comprising impairments (n = 35, 26%), performance (n = 77, 58%) and activity in a natural environment (n = 22, 16%; S4 and S5 Appendices). Physical measures (specific measure of a physical assessment, e.g., lumbar flexion range of movement) were categorized into 17 physical outcome measures (broad outcome that is measured by physical assessment, e.g., range of movement) and mapped to 15 level-two categories in ICF components Body Function and Activities/Participation [15]:

ICF Body Function component

- Mobility of joint functions (physical outcome measure: range of movement)

- Muscle power functions (physical outcome measure: strength)

- Control of voluntary movement functions (physical outcome measure: motor control)

- Gait pattern functions (physical outcome measure: gait parameters)

- Exercise tolerance functions (physical outcome measure: aerobic capacity)

- Muscle endurance functions (physical outcome measure: muscle endurance)

- Involuntary movement reaction functions (physical outcome measure: balance)

  ICF Activities/Participation component

- Changing basic body position (physical outcome measure: functional mobility)

- Maintaining body position (physical outcome measure: sustained positions)

- Lifting and carrying objects (physical outcome measure: lifting)

- Hand and arm use (physical outcome measure: reaching)

- Walking (physical outcome measures: walking, composite gait measure)

- Going up and down stairs (physical outcome measure: stairs)

- Moving around in different locations (physical outcome measure: walking)

- Other specified mobility (physical outcome measures: multi-activity performance-based measures, functional task performance, physical activity parameters)

The most frequently used physical outcome measure was range of movement for impairment-based, walking tests for performance-based and physical activity parameters for activity in a natural environment. Similar to PROMs, the importance of walking is highlighted by 21 different walking physical measures (e.g., 6-minute walk test) across performance-based and activity in a natural environment physical outcome measures.

## Stage two: Assess measurement properties of physical measures

**Study characteristics.**   Stage two included 43 articles (Table 1) evaluating measurement properties of 34 physical measures of impairments (n = 8), performance (n = 18) and activity in a natural environment (n = 8). Studies were published across 13 countries over 24 years. The total number of participants was 4,619 and sample size ranged from 8–375 (median = 50). Mean age of participants was 57 (range: 27–91). Reporting of sex versus gender was variable (70% sex, 30% gender) and binary with slightly more men/males (54%). Surgeries included discectomy, decompression and fusion for lumbar disc herniation, spinal stenosis, degenerative disc disease and spondylolisthesis.

**RoB in individual studies.**   Overall RoB in individual studies was rated as inadequate (78%, n = 76), doubtful (18%, n = 17) and very good (4%, n = 4; Tables 2–4). Key issues included lack of a priori hypotheses for hypothesis testing approaches and reporting measurement properties for comparator instruments (S6 Appendix). Complete agreement in RoB assessment was achieved through discussion (κ = 0.81).

**Measurement property results per physical measure.**   Investigations included assessments of validity (n = 22 physical measures; Table 2), responsiveness (n = 20 physical measures; Table 3), reliability (n = 8 physical measures; Table 4) and measurement error (n = 6 physical measures; Table 4). The strongest evidence was moderate-level, supporting sufficient responsiveness of 1-minute stair climb and 50-foot walk tests, insufficient responsiveness of the 5-minute walk test, and sufficient reliability of distance walked during the 6-minute walk test. Very low to low-level evidence limits further understanding of measurement properties. Measurement properties for all physical measures are summarized in Table 5. S6 Appendix details individual studies of measurement properties per physical measure and overall quality of evidence.

*Impairment-based physical outcome measures. Active range of movement*: very low-level evidence supports indeterminant construct validity and responsiveness (construct approach) of computer assisted electronic inclinometer measures of lumbar, trunk and hip flexion and extension, and dual bubble inclinometer measures of lumbopelvic flexion and extension [40, 44, 45]. Very low-level evidence supports indeterminant responsiveness (construct approach) of the Schober test[40].

*Handgrip strength*: low-level evidence supports indeterminant construct validity of maximum handgrip strength using a handgrip dynamometer [41–43].

*Gait parameters*: low level evidence supports indeterminant construct validity and sufficient criterion validity of the Two-step test [39]. Very low-level evidence supports indeterminant responsiveness (construct approach) for asymmetry of double support and stride length using a wireless gait analysis system [31].

*Performance-based physical outcome measures. 1-minute stair climb*: moderate-level evidence supports sufficient responsiveness (construct approach) of the 1-minute stair climb test [53].

*5 repetitions sit to stand*: low-level evidence supports sufficient reliability of time to complete the 5 repetitions sit-to-stand test [60, 61]. Low-level evidence supports indeterminant construct validity and measurement error of time to complete the 5 repetition sit-to-stand test [54, 57, 60–62].

*5-minute walk test*: moderate-level evidence supports insufficient responsiveness (construct approach) of the 5-minute walk test [53].

*6-minute walk test*: for distance walked during 6-minute walk test measured using the 6WT app [38, 55, 56, 59, 70] (measurement instrument not reported) [66, 67] moderate-level evidence supports sufficient test-retest reliability [38, 55], low-level evidence supports sufficient

**Table 1. Study and participant characteristics in stage two.**

| Author (Year) Country | Physical outcome measure: Physical measure | Study design | Sample size (n) | Gender / Sex, as reported (n) | Age (mean (SD)) | Spinal surgery and condition | Measurement property |
|---|---|---|---|---|---|---|---|
| **Impairment-based physical outcome measures** | | | | | | | |
| Fujita et al (2019) [39] Japan | Gait parameter: Two-step test | Prospective cross-sectional study | 357 | F: 156, M: 201 | 73 (5) | Primary surgery for LSS | Construct and criterion validity |
| Häkkinen et al (2005) [40] Finland | Active range of movement: Lumbar extension, Schober test | Prospective study (assessed as longitudinal observational) | 145 | F: 65, M: 80 | 41 (12) | Surgery for LDH | Responsiveness |
| Inoue et al (2020) [41] Japan | Handgrip strength: Handgrip MVC | Retrospective study (assessed as cross-sectional) | 183 | F: 55, M: 128 | 71, Range: 36–88 | Laminectomy for LSS | Construct validity |
| Kwon et al (2020) [42] South Korea | Handgrip strength: Handgrip MVC | Retrospective study (assessed as longitudinal observational) | 91 | W: 48, M: 43 | 68, Range: 33–86 | Decompression and fusion for LSS | Construct validity |
| Kwon et al (2020) [43] South Korea | Handgrip strength: Handgrip MVC | Retrospective observational study (assessed as longitudinal) | 200 | W: 126, M: 74 | Women: High grip strength: 65 (7); Low grip strength: 70 (7) Men: High grip strength: 65 (10); Low grip strength: 70 (7) | Decompression and fusion for LSS | Construct validity |
| Loske et al (2018) [31] Switzerland | Gait parameters: Asymmetry of double support, Stride length | Prospective observational study with intervention (assessed as longitudinal with case control) | Pre-op: 35, 10 weeks: 29, 12 months: 20 | F: 12, M: 17 (10 week data reported) | Range: 58–83 | Decompression with or without fusion for LSS | Responsiveness |
| Mannion et al (2005) [44] Switzerland | Active range of movement: Lumbar, trunk and hip flexion and extension | Prospective study (assessed as longitudinal observational with case control) | 33 | W: 9, M: 24 | 57 (9) | Decompression for LDH | Construct validity, Responsiveness |
| Pitino (2000) [45] USA | Active range of movement: Lumbopelvic flexion and extension | Prospective cohort (assessed as longitudinal observational) | 16 | F: 6, M: 10 | 36 (6), Range: 27–45 | Discectomy for LDH | Construct validity |
| **Performance-based physical outcome measures** | | | | | | | |
| Corniola et al (2016) [46] Switzerland | Timed Up and Go: Time to complete | Prospective study (assessed as observational cross-sectional) | 284 | F: 119, M: 165 | 59 (16) | Microdiscectomy, decompression or fusion for LDH, LSS or DDD with or without instability | Construct validity |
| Dedering (2006) [47] Sweden | Modified Sørensen: Time to exhaustion | Prospective study (assessed as longitudinal observational) | 43 | W: 16, M: 27 | 42 (11) | Microdiscectomy for LDH | Construct validity |
| Dedering et al (2012) [48] Sweden | Modified Sørensen: Time to exhaustion | Prospective study (assessed as longitudinal observational) | 26 | W: 7, M: 19 | 42 (11) | Microdiscectomy for LDH | Construct validity |

(*Continued*)

**Table 1.** (Continued)

| Author (Year) Country | Physical outcome measure: Physical measure | Study design | Sample size (n) | Gender / Sex, as reported (n) | Age (mean (SD)) | Spinal surgery and condition | Measurement property |
|---|---|---|---|---|---|---|---|
| Deen et al (2000) [49] United States | Treadmill test: Time to first symptoms, Total ambulation time | Prospective study (assessed as longitudinal observational) | Pre-operative: 28 Post-operative: 18 | Pre-operative: W: 11, M: 17 Post-operative: NR | 74, Range: 57–91 | Laminectomy for stenosis | Test-retest reliability |
| Gautschi et al (2016) [50] Switzerland | Timed Up and Go: Time to complete | Prospective study (assessed as cross-sectional with case control) | 253 | F: 107, M: 146 | 59 (16) | Surgery for LDH, LSS or DDD with or without instability | Construct validity |
| Gautschi et al (2016) [51] Switzerland | Timed Up and Go: Time to complete | Prospective study (assessed as longitudinal observational) | 136 | F: 60, M: 76 | 58 (16) | Microdiscectomy, decompression or fusion for LDH, LSS or DDD with or without instability | Responsiveness |
| Häkkinen et al (2005) [40] Finland | Trunk muscle endurance: Repetitive arch ups and sit ups until exhaustion | Prospective study (assessed as longitudinal observational) | 145 | F: 65, M: 80 | 41 (12) | Surgery for LDH | Responsiveness |
| Herno et al (1999) [52] Finland | Treadmill test: Maximum walking distance | Retrospective cross sectional | 56 | W: 25, M: 31 | W: 55, M: 54 | Laminectomy for LSS | Construct validity |
| Jakobsson et al (2019) [53] Sweden | 1-min stair climb: Number of stairs 5-min walk: Distance walked 50-ft walk: Time to complete Timed Up and Go: Time to complete | Prospective design using data from randomized controlled trial | 93 | W: 51, M: 42 | 47 (8) | Fusion for motion-provoked chronic LBP with degenerative changes | Responsiveness |
| Klukowska et al (2020) [54] Netherlands | 5 repetition sit to stand test: Time to complete | Used data from two prospective cohort studies (assessed as cross-sectional) | 240 | F: 115, M: 125 | No OFI: 48 (15) Mild OFI: 51 (13) Moderate OFI: 43 (12) Severe OFI: 44 (11) | Surgery for LDH, LSS, DDD, Spondylolisthesis | Construct validity |
| Maldaner et al (2020) [55] Switzerland | 6-min walk test: Distance walked | Prospective study (assessed as cross-sectional with case control) | 70 | F: 27, M: 43 | 56 (15) | Surgery for LDH, LSS, DDD | Construct validity, Measurement error, Test-retest reliability |
| Maldaner et al (2021) [56] Switzerland | 6-min walk test: Distance walked Timed Up and Go: Time to complete | Prospective study (assessed as longitudinal observational) | 49 | F: 20, M: 29 | 56 (16) | Microdiscectomy for LDH, Decompression for LSS, Fusion for DLD with or without instability | Construct validity, Responsiveness |
| Master et al (2020) [57] United States | 5 repetition sit to stand test: Time to complete | Secondary analysis of randomized controlled trial | 248 | F: 126, M: NR | 62 (12) | Laminectomy with or without fusion for spondylosis, degenerative spondylolisthesis and spinal stenosis | Construct validity |
| Prasad et al (2016) [58] India | Treadmill test: Maximum walking distance, Maximum walking time, Distance to first symptoms, Time to first symptoms | Prospective study (assessed as longitudinal observational) | 48 | F: 26, M: 22 | Age at inception into study NR. Age at development of lumbar canal stenosis: 46 (8) Eligibility: 18–65 years | Decompression with or without fusion for lumbar canal stenosis | Construct validity |

(*Continued*)

**Table 1.** (*Continued*)

| Author (Year) Country | Physical outcome measure: Physical measure | Study design | Sample size (n) | Gender / Sex, as reported (n) | Age (mean (SD)) | Spinal surgery and condition | Measurement property |
|---|---|---|---|---|---|---|---|
| Sosnova et al (2021) [59] Switzerland | 6-min walk test: Distance walked | Prospective observational cohort study (assessed as longitudinal) | 49 | F: 20, M: 29 | 56 (16) | Surgery for LDH, LSS, or DDD with or without instability | Responsiveness |
| Staartjes et al (2018) [60] Switzerland | 5 repetition sit to stand test: Time to complete | Prospective study (assessed as case-controlled cross-sectional) | 157 | Male: 80 | 50 (14) | Surgery for LDH, LSS, DDD, lumbar spondylolisthesis or synovial facet cysts | Measurement error, Test-retest reliability |
| Staartjes et al (2019) [61] Switzerland | 5 repetition sit to stand test: Time to complete | Prospective study (assessed as cross-sectional) | 100 | 'Male gender': 44 | 45 (12) | Surgery for LDH, LSS, DDD or spondylolisthesis | Construct validity, Measurement error, Inter-rater and test-retest reliability |
| Staartjes et al (2020) [62] Netherlands | 5 repetition sit to stand test: Time to complete | Pooled prospectively collected data from two cohort studies (assessed as cross-sectional) | 258 | F: 126, M: 132 | 48 (14) | Surgery for LDH, LSS, DDD or spondylolisthesis | Construct validity |
| Staartjes et al (2022) [63] Switzerland | Timed Up and Go: Time to complete | Secondary analysis of prospective database (assessed as cross-sectional) | 375 | F: 162 (43%) | 59 (16) | Discectomy, decompression, fusion for LDH, LSS or lumbar DDD requiring surgical treatment | Construct validity |
| Stienen et al (2017) [64] Switzerland and Australia | Timed Up and Go: Time to complete | Prospective observational study (assessed as cross-sectional) | 375 | No OFI: F: 100, M: 130 Mild OFI: F: 24, M: 25 Moderate OFI: F: 25, M: 30 Severe OFI: F: 13, M: 28 | No OFI 62 (15) Mild OFI 52 (16) Moderate OFI 57 (16) Severe OFI 52 (16) | Microdiscectomy, decompression, fusion for LDH, LSS, DDD with or without instability | Construct validity |
| Stienen et al (2021) [65] Switzerland | Timed Up and Go: Time to complete | Prospective observational study (assessed as cross-sectional) | 70 | F: 27, M: 43 | 56 (15) | Microdiscectomy for LDH, decompression for LSS, fusion for DDD with or without instability | Construct validity |
| Takenaka et al (2022) [66] Japan | 6-min walk test: Distance walked | Retrospective study (assessed as longitudinal observational) | 41 | W: 16, M: 25 | 69 (8) | Fusion or decompression for LSS | Responsiveness |
| Takenaka et al (2023) [67] Japan | 6-min walk test: Distance walked | Retrospective study (assessed as longitudinal observational) | 126 | W: 52, M: 74 reported as sex | 72 (7) | Fusion and / or decompression for LSS | Responsiveness, Measurement error |
| Tomkins-Lane et al (2020) [68] Canada | Self-paced walking test: Distance walked, Time walked | Prospective pilot cohort study (assessed as longitudinal observational) | 21 | F: 15, M: 6 | 65 (10), Median: 66, Range 44–79 | Decompression for LSS | Responsiveness |

(*Continued*)

**Table 1.** (Continued)

| Author (Year) Country | Physical outcome measure: Physical measure | Study design | Sample size (n) | Gender / Sex, as reported (n) | Age (mean (SD)) | Spinal surgery and condition | Measurement property |
|---|---|---|---|---|---|---|---|
| Wada et al (2022) [69] Japan | 10-meter walk test: Walking speed | Prospective observational (assessed as longitudinal) | Pre-operative: 94 3 months post-operative: 83 6 months post-operative: 88 12 months post-operative: 82 | F: 49% | 70 (9) | Decompression with or without fusion for LSS | Construct validity, Responsiveness |
| Zeitlberger et al (2022) [38] Switzerland | 6-min walk test: Distance walked, Distance to first symptoms, Time to first symptoms | Prospective study (assessed as longitudinal observational) | 49 | F: 20, M: 29 | 56 (16) | Microdiscectomy, decompression or fusion for LDH, LSS or DLD with or without instability | Construct validity, Measurement error, Test-retest reliability, Responsiveness |
| Ziga et al (2023) [70] Switzerland | 6-min walk test: Distance walked | Prospective observational cohort study | 50 | F: 19, M: 31 | 58 (14) | Microdiscectomy, decompression or fusion for LDH, LSS or DLD with or without instability | Construct validity, Responsiveness |
| **Activity in a natural environment physical outcome measures** | | | | | | | |
| Bienstock et al (2022) [71] USA | Steps per day | Prospective cohort study (assessed as longitudinal observational) | Pre-operative: 20 Post-operative 2 weeks: 17 1 month: 18 3 months: 16 6 months: 14 | F: 9, M: 11 (pre-operative) | 65 (9), Range 50–79 | Laminectomy with or without 1–2 level fusion for LSS with or without spondylolisthesis | Construct validity |
| Ghent et al (2020) [72] Australia | Gait Posture Index | Prospective series (assessed as longitudinal observational) | 24 | W: 10, M: 14 | 50 (15), Range: 51–83 | Microdiscectomy for LDH | Responsiveness |
| Gilmore et al (2020) [73] Australia | Step counts | Not reported (assessed as cross-sectional) | 40 | F: 26, M: 14 | 65 (13) | 1 level fusion for DDD, LSS and / or degenerative spondylolisthesis | Criterion validity, Measurement error |
| Kim et al (2019) [74] South Korea | Steps per day | Prospective study (assessed as longitudinal observational) | 22 | F: 11, M: 11 | 60 (13) | 1 level partial laminectomy for LSS or LDH | Construct validity |
| Mobbs et al (2016) [75] Australia | Steps per day, Distance per day | Prospective observational study (assessed as longitudinal) | 28 | Men: 17 | 43 (10) | Fusion, laminectomy, or discectomy for LBP, radiculopathy and / or claudication | Responsiveness |
| Mobbs et al (2019) [76] Australia | Gait Posture Index | Prospective non-randomized series (assessed as longitudinal observational) | 13 | F: 11, M: 2 | 69 (11), Range: 51–83 | Decompression for LSS | Responsiveness |

(*Continued*)

**Table 1.** (Continued)

| Author (Year) Country | Physical outcome measure: Physical measure | Study design | Sample size (n) | Gender / Sex, as reported (n) | Age (mean (SD)) | Spinal surgery and condition | Measurement property |
|---|---|---|---|---|---|---|---|
| Schulte et al (2010) [77] Germany | Gait cycles per day, Gait cycles per hour, Gait intensities per day | Prospective, controlled study (assessed as longitudinal observational) | 47 | W: 24, M: 23 | 69 (8) | Decompression with or without fusion for central LSS | Construct validity |
| Stienen et al (2020) [78] USA | Steps per day | Prospective observational study (assessed as longitudinal) | Study sample: 30 Lumbar cohort: 18 | Study sample: F: 15, M: 15 Lumbar cohort: NR | Study sample: 57 (15) Lumbar cohort: NR (Eligibility criteria >18 years old) | Discectomy / decompression with or without instrumentation / fusion for LDH, LSS, or spondylolisthesis | Construct validity |
| Voglis et al (2022) [79] Switzerland | Distance walked per day | Prospective observational study | 8 | F: 3, M: 5 | 46 (13) | Microdiscectomy, decompression, or fusion for LDH, LSS, DLD | Construct validity |

DDD, Degenerative disc disease; DLD, Degenerative lumbar disorders; F, Female; LDH, Lumbar disc herniation; LSS, Lumbar spinal stenosis; M, Male / Man; MVC, Maximum voluntary contraction; NR, Not reported; OFI, Objective functional impairment W, Woman.

responsiveness (criterion approach), inconsistent measurement error, and indeterminant construct validity and responsiveness (construct approach) [38, 55, 56, 66, 67, 70], and very low-level evidence supports patient reported responsiveness [59]. For distance to first symptoms during the 6-minute walk test measured using the 6WT app, very low-level evidence supports sufficient test-retest reliability and responsiveness (criterion approach) [38], and indeterminant construct validity, responsiveness (construct approach) and measurement error [38]. For time to first symptoms during the 6-minute walk test measured using the 6WT app, very low-level evidence supports sufficient test-retest reliability, insufficient responsiveness (criterion approach) [38], and indeterminant construct validity, responsiveness (construct approach) and measurement error [38].

*10-meter walk test*: very low-level evidence supports indeterminant construct validity and responsiveness (construct approach) of the 10-meter walk test [69].

*50-foot walk test*: moderate-level evidence supports sufficient responsiveness (construct approach) of the 50-foot walk test [53]

*Modified Sorensen test*: very low-level evidence supports indeterminant construct validity of the modified Sorensen test [47, 48].

*Self-paced walking test*: very low-level evidence supports indeterminant responsiveness (construct approach) of distance and time walked during the self-paced walking test [68].

*Timed up and go*: low-level evidence supports inconsistent responsiveness (construct approach) of time to complete the TUG using the 6WT app [56] or stopwatch [51] (not reported) [53]. Very low-level evidence supports insufficient responsiveness (criterion approach) and indeterminant construct validity using the 6WT app [56, 63, 65] or stopwatch [64] (not reported) [46, 50, 57].

*Treadmill test*: for maximum walking time and time to first symptoms during a treadmill test, very low-level evidence supports sufficient pre-operative test-retest reliability and indeterminant post-operative test-retest reliability [49]. Low to very low-level evidence supports indeterminant construct validity for maximum walking distance, maximum walking time, time to first symptoms and distance to first symptoms during a treadmill test [52, 58].

*Trunk muscle endurance*: very low-level evidence supports indeterminant responsiveness (construct approach) of repetitive arch ups and sit ups until exhaustion [40].

**Table 2. Summary of findings—Validity.**

| Physical outcome measure / physical measure | Measurement instrument | Study | Construct validity | | | | Criterion validity | | | |
|---|---|---|---|---|---|---|---|---|---|---|
| | | | Result (Rating) | Risk of bias | Overall rating [a] | Overall quality of evidence | Result (Rating) | Risk of bias | Overall rating [a] | Overall quality of evidence |
| **Impairment-based physical outcome measures** | | | | | | | | | | |
| **Physical outcome measure: Active range of movement** | | | | | | | | | | |
| Physical measure: Lumbar, trunk and hip flexion and extension | Computer assisted electronic goniometer | Mannion et al (2005) [44] | No hypothesis defined (?) | Inadequate | ? | Very low | Not studied | | | |
| Physical measure: Lumbopelvic flexion and extension | Dual bubble inclinometers | Pitino (2000) [45] | No hypothesis defined (?) | Inadequate | ? | Very low | Not studied | | | |
| **Physical outcome measure: Handgrip strength** | | | | | | | | | | |
| Physical measure: Maximum voluntary contraction | Handheld dynamometer | Kwon et al (2020) [42] | No hypothesis defined (?) | Inadequate | ? | Low | Not studied | | | |
| | | Kwon et al (2020) [43] | No hypothesis defined (?) | Inadequate | | | | | | |
| | | Inoue et al (2020) [41] | No hypothesis defined (?) | Inadequate | | | | | | |
| **Physical outcome measure: Gait parameters** | | | | | | | | | | |
| Physical measure: Two-test test | NR | Fujita et al (2019) [39] | No hypothesis defined (?) | Inadequate | ? | Low | AUC: 0.83 (95% CI 0.78–0.89), (+) | Doubtful | + | Low |
| **Performance-based physical outcome measures** | | | | | | | | | | |
| **Physical outcome measure: 5 repetitions sit to stand** | | | | | | | | | | |
| Physical measure: Time to complete | NR | Master et al (2020) [57] | No hypothesis defined (?) | Inadequate | ? | Low | Not studied | | | |
| | Clinic, Unsupervised tests: NR Tele-supervised: Digital timer | Staartjes et al (2019) [61] | No hypothesis defined (?) | Inadequate | | | | | | |
| | Stopwatch | Staartjes et al (2020) [62] | No hypothesis defined (?) | Inadequate | | | | | | |
| | | Klukowska et al (2020) [54] | No hypothesis defined (?) | Inadequate | | | | | | |
| | | Staartjes et al (2018) [60] | No hypothesis defined (?) | Inadequate | | | | | | |
| **Physical outcome measure: 6-minute walk test** | | | | | | | | | | |

*(Continued)*

**Table 2.** (Continued)

| Physical outcome measure / physical measure | Measurement instrument | Study | Construct validity | | | | Criterion validity | | | |
|---|---|---|---|---|---|---|---|---|---|---|
| | | | Result (Rating) | Risk of bias | Overall rating [a] | Overall quality of evidence | Result (Rating) | Risk of bias | Overall rating [a] | Overall quality of evidence |
| Physical measure: Distance walked | 6WT app | Maldaner et al (2020) [55] | No hypothesis defined (?) | Inadequate | ? | Low | Not studied | | | |
| | | Maldaner et al (2021) [56] | No hypothesis defined (?) | Inadequate | | | | | | |
| | | Zeitlberger et al (2022) [38] | No hypothesis defined (?) | Inadequate | | | | | | |
| | | Ziga et al (2023) [70] | No hypothesis defined (?) | Inadequate | | | | | | |
| Physical measure: Distance to first symptoms | 6WT app | Zeitlberger et al (2022) [38] | No hypothesis defined (?) | Inadequate | ? | Very low | Not studied | | | |
| Physical measure: Time to first symptoms | 6WT app | Zeitlberger et al (2022) [38] | No hypothesis defined (?) | Inadequate | ? | Very low | Not studied | | | |
| **Physical outcome measure: 10-meter walk test** | | | | | | | | | | |
| Physical measure: Walking speed | NR | Wada et al (2022) [69] | No hypothesis defined (?) | Inadequate | ? | Very low | Not studied | | | |
| **Physical outcome measure: Modified Sorensen test** | | | | | | | | | | |
| Physical measure: Time to exhaustion | NR | Dedering et al (2006) [47] | No hypothesis defined (?) | Inadequate | ? | Very low | Not studied | | | |
| | | Dedering (2012) [48] | No hypothesis defined (?) | Inadequate | | | | | | |
| **Physical outcome measure: Timed Up and Go** | | | | | | | | | | |
| Physical measure: Time to complete | TUG app | Maldaner et al (2021) [56] | No hypothesis defined (?) | Inadequate | ? | Very low | Not studied | | | |
| | | Stienen et al (2021) [65] | 9/9 hypotheses confirmed (+) | Doubtful | | | | | | |
| | | Staartjes et al (2022) [63] | No hypothesis defined (?) | Inadequate | | | | | | |
| | NR | Master et al (2020) [57] | No hypothesis defined (?) | Inadequate | | | | | | |
| | | Corniola et al (2016) [46] | No hypothesis defined (?) | Inadequate | | | | | | |
| | | Gautschi et al (2016) [50] | No hypothesis defined (?) | Inadequate | | | | | | |
| | Stopwatch | Stienen et al (2017) [64] | No hypothesis defined (?) | Inadequate | | | | | | |
| **Physical outcome measure: Treadmill test** | | | | | | | | | | |

*(Continued)*

**Table 2.** (Continued)

| Physical outcome measure / physical measure | Measurement instrument | Study | Construct validity | | | | Criterion validity | | | |
|---|---|---|---|---|---|---|---|---|---|---|
| | | | Result (Rating) | Risk of bias | Overall rating [a] | Overall quality of evidence | Result (Rating) | Risk of bias | Overall rating [a] | Overall quality of evidence |
| Physical measure: Maximum walking distance | Treadmill, 3.6 km/hr | Herno et al (1999) [52] | No hypothesis defined (?) | Inadequate | ? | Low | Not studied | | | |
| | Treadmill, 2 km/hr, 0% incline | Prasad et al (2016) [58] | No hypothesis defined (?) | Inadequate | | | | | | |
| Physical measure: Maximum walking time | Treadmill, 2 km/hr, 0% incline | Prasad et al (2016) [58] | No hypothesis defined (?) | Inadequate | ? | Very low | Not studied | | | |
| Physical measure: Time to first symptoms | Treadmill, 2 km/hr, 0% incline | Prasad et al (2016) [58] | No hypothesis defined (?) | Inadequate | ? | Very low | Not studied | | | |
| Physical measure: Distance to first symptoms | Treadmill, 2 km/hr, 0% incline | Prasad et al (2016) [58] | No hypothesis defined (?) | Inadequate | ? | Very low | Not studied | | | |
| **Activity in a natural environment physical outcome measures** | | | | | | | | | | |
| **Physical outcome measure: Step counts** | | | | | | | | | | |
| Physical measure: Steps per day | Fitbit Charge | Kim et al (2019) [74] | No hypothesis defined (?) | Inadequate | ? | Very low | Not studied | | | |
| | Mi Band | Stienen et al (2020) [78] | No hypothesis defined (?) | Inadequate | | | | | | |
| | Fitbit Flex 2 | Bienstock et al (2022) [71] | No hypothesis defined (?) | Inadequate | | | | | | |
| Physical measure: Steps detected at thigh | ActivPAL3 | Gilmore et al (2020) [73] | Not studied | | | | ICC: 0.81 (95% CI 0.37–0.94), (+) | Doubtful | ± | Very low |
| | Jawbone UP Move | Gilmore et al (2020) [73] | | | | | ICC: 0.71 (95% CI -0.02–0.91), (+) | | | |
| | Fitbit Flex | Gilmore et al (2020) [73] | | | | | ICC: 0.11 (95% CI -0.15–0.44), (-) | | | |

**Table 2.** (Continued)

| Physical outcome measure / physical measure | Measurement instrument | Study | Construct validity | | | | Criterion validity | | | |
|---|---|---|---|---|---|---|---|---|---|---|
| | | | Result (Rating) | Risk of bias | Overall rating [a] | Overall quality of evidence | Result (Rating) | Risk of bias | Overall rating [a] | Overall quality of evidence |
| Physical measure: Steps detected at wrist | Jawbone UP Move | Gilmore et al (2020) [73] | Not studied | | | | Total: ICC: 0.36 (95% CI -0.17–0.74), (-) No gait aid: ICC: 0.46 (95% CI -0.36–0.87), (-) With gait aid: Unable to calculate, no steps detected (-) | Doubtful | - | Very low |
| | Fitbit Flex | Gilmore et al (2020) [73] | | | | | Total: ICC: 0.35 (95% CI -0.17–0.74), (-) No gait aid: ICC: 0.36 (95% CI -0.23–0.79), (-) With gait aid: ICC: 0.13, (95% CI -0.10–0.55), (-) | | | |
| **Physical outcome measure: Distance walked per day** | | | | | | | | | | |
| Physical measure: Distance walked per day | Personal smart phone Apple Health data | Voglis et al (2022) [79] | No hypothesis defined (?) | | Inadequate | ? | Very low | Not studied | | | |
| **Physical outcome measure: Gait cycles** | | | | | | | | | | |
| Physical measure: Gait cycles per day | StepWatch 3 Activity Monitor | Schulte et al (2010) [77] | No hypothesis defined (?) | | Inadequate | ? | Very low | Not studied | | | |
| Physical measure: Gait cycles per hour | StepWatch 3 Activity Monitor | Schulte et al (2010) [77] | No hypothesis defined (?) | | Inadequate | ? | Very low | Not studied | | | |
| Physical measure: Gait intensities per day | StepWatch 3 Activity Monitor | Schulte et al (2010) [77] | No hypothesis defined (?) | | Inadequate | ? | Very low | Not studied | | | |

[a] Rating according to COSMIN criteria for good measurement properties: (+) Sufficient; (-) Insufficient; (?) Indeterminant; (±) Inconsistent.

AUC, Area under the curve; CI, Confidence interval; ICC, intraclass correlation coefficient; km/hr, kilometers per hour; NR, not reported.

*Activity in a natural environment physical outcome measures. Step counts*: very low-level evidence supports indeterminant construct validity and responsiveness (construct approach) of steps per day using a Fitbit (Flex 2, Charge, Zip) [71, 74, 75] and Mi Band [78]. At the thigh, very low-level evidence supports inconsistent criterion validity of step detection (sufficient for ActivPAL3 and Jawbone UP Move, insufficient for Fitbit Flex) [73]. At the wrist, very low-level evidence supports insufficient criterion validity of step detection using a Fitbit Flex and Jawbone UP Move [73]. At the thigh and wrist, very low-level evidence supports indeterminant measurement error of step detection using Fitbit Flex, Jawbone UP Move and ActivPAL3 (thigh only) [73].

**Table 3. Summary of findings—Responsiveness.**

| Physical outcome measure / physical measure | Measurement instrument | Study | Responsiveness—Construct approach hypothesis testing | | | | Responsiveness—Criterion approach | | | |
|---|---|---|---|---|---|---|---|---|---|---|
| | | | Result (Rating) | Risk of bias | Overall rating [a] | Overall quality of evidence | Result (Rating) | Risk of bias | Overall rating [a] | Overall quality of evidence |
| **Impairment-based physical outcome measures** | | | | | | | | | | |
| **Physical outcome measure: Active Range of Movement** | | | | | | | | | | |
| Physical measure: Lumbar extension | Dualer goniometer | Häkkinen et al (2005) [40] | No hypothesis defined (?) | Inadequate | ? | Very low | Not studied | | | |
| Physical measure: Lumbar, trunk and hip flexion and extension | Computer assisted electronic goniometer | Mannion et al (2005) [44] | No hypothesis defined (?) | Inadequate | ? | Very low | Not studied | | | |
| Physical measure: Schober test | NR | Häkkinen et al (2005) [40] | No hypothesis defined (?) | Inadequate | ? | Very low | Not studied | | | |
| **Physical outcome measure: Gait parameters** | | | | | | | | | | |
| Physical measure: Asymmetry of double support | RehabGait | Loske et al (2018) [31] | No hypothesis defined (?) | Inadequate | ? | Very low | Not studied | | | |
| Physical measure: Stride length | RehabGait | Loske et al (2018) [31] | No hypothesis defined (?) | Inadequate | ? | Very low | Not studied | | | |
| **Performance-based physical outcome measures** | | | | | | | | | | |
| **Physical outcome measure: 1-minute stair climb** | | | | | | | | | | |
| Physical measure: Number of stairs | Observer counts steps on 10-step staircase | Jakobsson et al (2019) [53] | 4 / 5 hypotheses confirmed (+) | Very good | + | Moderate | Not studied $Note$: Jakobsson et al (2019) investigated $AUC_{Construct-specific\ GPE}$ and $AUC_{Generic\ GPE}$ but within context of hypothesis testing | | | |
| **Physical outcome measure: 5-minute walk test** | | | | | | | | | | |
| Physical measure: Distance walked | 30m long octagonal circuit, measurement instrument NR | Jakobsson et al (2019) [53] | 2 / 5 hypotheses confirmed (-) | Very good | - | Moderate | Not studied $Note$: Jakobsson et al (2019) investigated $AUC_{Construct-specific\ GPE}$ and $AUC_{Generic\ GPE}$ but within context of hypothesis testing | | | |
| **Physical outcome measure: 6-minute walk test** | | | | | | | | | | |

(*Continued*)

**Table 3.** (Continued)

| Physical outcome measure / physical measure | Measurement instrument | Study | Responsiveness—Construct approach hypothesis testing | | | | Responsiveness—Criterion approach | | | |
|---|---|---|---|---|---|---|---|---|---|---|
| | | | Result (Rating) | Risk of bias | Overall rating [a] | Overall quality of evidence | Result (Rating) | Risk of bias | Overall rating [a] | Overall quality of evidence |
| Physical measure: Distance walked | 30m path, measurement instrument NR | Takenaka et al (2022) [66] | No hypothesis defined (?) | Inadequate | ? | Low | AUC: 0.70 (95% CI 0.52–0.89), (+) | Doubtful | + | Low |
| | | Takenaka et al (2023) [67] | No hypothesis defined (?) | Inadequate | | | 6 month: AUC: 0.72 (95% CI 0.63–0.82), (+) 12 month: AUC: 0.78 (95% CI 0.69–0.86), (+) 6 month— Severe disability: AUC: 0.98 (95% CI 0.94–1.00), (+) 12 month– Severe disability: AUC: 0.90 (95% CI 0.68–1.00), (+) | Doubtful | | |
| | | | | | | | 6 month–Low disability: AUC: 0.60 (95% CI 0.43–0.76), (-) 12 month–Low disability: AUC: 0.51 (95% CI 0.34–0.67), (-) | Doubtful | | |
| | 6WT app | Zeitlberger et al (2022) [38] | No hypothesis defined (?) | Inadequate | | | AUC: 0.70 (95% CI: 0.52–0.90), (+) | Inadequate | | |
| | | Maldaner et al (2021) [56] | No hypothesis defined (?) | Inadequate | | | AUC: 0.70 (95% CI: 0.51–0.89), (+) | Doubtful | | |
| | | Ziga et al (2023) [70] | No hypothesis defined (?) | Inadequate | | | Not studied | | | |
| | Patient reported responsiveness (Survey question following use of 6WT app) | Sosnova et al (2021) [59] | No hypothesis defined (?) | Inadequate | ? | Very low | Not studied | | | |
| Physical measure: Distance to first symptoms | 6WT app | Zeitlberger et al (2022) [38] | No hypothesis defined (?) | Inadequate | ? | Very low | AUC: 0.75 (95% CI: 0.53–0.98), (+) | Inadequate | + | Very low |
| Physical measure: Time to first symptoms | 6WT app | Zeitlberger et al (2022) [38] | No hypothesis defined (?) | Inadequate | ? | Very low | AUC: 0.59 (95% CI: 0.34–0.83), (-) | Inadequate | - | Very low |
| **Physical outcome measure: 10-meter walk test** | | | | | | | | | | |

(*Continued*)

**Table 3.** (Continued)

| Physical outcome measure / physical measure | Measurement instrument | Study | Responsiveness—Construct approach hypothesis testing | | | | Responsiveness—Criterion approach | | | |
|---|---|---|---|---|---|---|---|---|---|---|
| | | | Result (Rating) | Risk of bias | Overall rating [a] | Overall quality of evidence | Result (Rating) | Risk of bias | Overall rating [a] | Overall quality of evidence |
| Physical measure: Walking speed | NR | Wada et al (2022) [69] | No hypothesis defined (?) | Inadequate | ? | Very low | Not studied | | | |
| **Physical outcome measure: 50-foot walk test** | | | | | | | | | | |
| Physical measure: Time to complete | 15 m circuit, figure 8 shaped, measurement instrument NR | Jakobsson et al (2019) [53] | 4 / 5 hypotheses confirmed (+) | Very good | + | Moderate | Not studied Note: Jakobsson et al (2019) investigated $AUC_{Construct-specific\ GPE}$ and $AUC_{Generic\ GPE}$ but within context of hypothesis testing | | | |
| **Physical outcome measure: Self-paced walking test** | | | | | | | | | | |
| Physical measure: Distance walked | NR | Tomkins-Lane et al (2020) [68] | No hypothesis defined (?) | Inadequate | ? | Very low | Not studied | | | |
| Physical measure: Time walked | NR | Tomkins-Lane et al (2020) [68] | No hypothesis defined (?) | Inadequate | ? | Very low | Not studied | | | |
| **Physical outcome measure: Timed Up and Go** | | | | | | | | | | |
| Physical measure: Time to complete | 6WT app | Maldaner et al (2021) [56] | No hypothesis defined (?) | Inadequate | ± | Low | AUC: 0.53 (95% CI: 0.30–0.77), (-) | Doubtful | - | Very low |
| | NR | Jakobsson et al (2019) [53] | 5 / 5 hypotheses confirmed (+) | Very good | | | Not studied Note: Jakobsson et al (2019) investigated $AUC_{Construct-specific\ GPE}$ and $AUC_{Generic\ GPE}$ but within context of hypothesis testing | | | |
| | Stopwatch | Gautschi (2016) [51] | No hypothesis defined (?) | Inadequate | | | Not studied | | | |
| **Physical outcome measure: Trunk muscle endurance** | | | | | | | | | | |
| Physical measure: Repetitive arch-ups until exhaustion | Observer counts number of repetitions | Häkkinen et al (2005) [40] | No hypothesis defined (?) | Inadequate | ? | Very low | Not studied | | | |
| Physical measure: Repetitive sit-ups until exhaustion | Observer counts number of repetitions | Häkkinen et al (2005) [40] | No hypothesis defined (?) | Inadequate | ? | Very low | Not studied | | | |
| **Activity in a natural environment physical outcome measures** | | | | | | | | | | |
| **Physical outcome measure: Step count** | | | | | | | | | | |
| Physical measure: Steps per day | Fitbit Zip | Mobbs et al (2016) [75] | No hypothesis defined (?) | Inadequate | ? | Very low | Not studied | | | |
| **Physical outcome measure: Gait Posture Index** | | | | | | | | | | |
| Physical measure: Gait Posture Index | Mi Band 2 or personal smart watch | Ghent et al (2020) [72] | No hypothesis defined (?) | Inadequate | ? | Very low | Not studied | | | |
| | | Mobbs et al (2019) [76] | No hypothesis defined (?) | Inadequate | | | | | | |
| **Physical outcome measure: Distance walked per day** | | | | | | | | | | |

*(Continued)*

**Table 3.** (Continued)

| Physical outcome measure / physical measure | Measurement instrument | Study | Responsiveness—Construct approach hypothesis testing | | | | Responsiveness—Criterion approach | | | |
|---|---|---|---|---|---|---|---|---|---|---|
| | | | Result (Rating) | Risk of bias | Overall rating [a] | Overall quality of evidence | Result (Rating) | Risk of bias | Overall rating [a] | Overall quality of evidence |
| Physical measure: Distance walked per day | Fitbit Zip | Mobbs et al (2016) [75] | No hypothesis defined (?) | Inadequate | ? | Very low | Not studied | | | |

[a] Rating according to COSMIN criteria for good measurement properties: (+) Sufficient; (-) Insufficient; (?) Indeterminant; (±) Inconsistent.

AUC, Area under the curve; CI, Confidence interval; ICC, intraclass correlation coefficient; NR, Not reported.

*Gait Posture Index*: very low-level evidence supports indeterminant responsiveness (construct approach) of the Gait Posture Index using personal electronic devices (e.g., Garmin) or Mi Band 2 [72, 76].

*Distance walked per day*: very low-level evidence supports indeterminant construct validity (using personal smart phone) [79] and responsiveness (construct approach, using Fitbit Zip) [75] of distance walked per day.

*Gait cycles*: very low-level evidence supports indeterminant construct validity of number of gait cycles per day, gait cycles per hour and gait intensities per day using the StepWatch3 [77].

**Interpretability and feasibility.** Limited information reported interpretability and feasibility (S6 Appendix). Floor and ceiling effects were generally not reported. Distribution of scores suggest floor effects in some physical measures with small scores (e.g., range of movement) [44, 45], while some walking physical measures may have floor (symptoms at test start) [49] and ceiling (no symptoms at test end) [38, 49, 68] effects. Data missingness was variable (0–69%). Minimum important or detectable change scores were rarely investigated in studies [53, 66, 67], though some values were reported from the literature [51, 56, 57, 64, 65]. Administering physical measures required small (e.g., accelerometer) or readily available equipment (e.g., goniometer), or a patient's personal electronic device (e.g., smartwatch), enabling relatively easy set up/administration within clinical environments. Most studies did not report equipment cost, but free smartphone applications exist [38, 55, 56, 59, 65]. Several digital technologies (personal electronic devices [38, 55, 56, 59, 61, 65, 72, 76], low-cost consumer-grade wearables [71–78]) may enhance feasibility of data collection because of minimal interference with daily activities. When reported, standardized instructions and ease of score calculation appear feasible in clinical environments.

**Reporting biases.** No study protocols of measurement properties were identified in stage two. Selective reporting of results was considered and reported within RoB assessment.

## Discussion

Using a rigorous two-staged approach, this systematic review is the first to identify outcome measures (PROMs, physical) used to evaluate physical functioning in the lumbar spinal surgery population and assess measurement properties of the physical measures. Stage one generated a comprehensive list of PROMs (Established n = 70, Developed n = 67) and physical measures (n = 134). However, only 34 physical measures had investigations of measurement properties. Moderate-level evidence supported sufficient responsiveness of the 1-minute stair climb and 50-foot walk tests, insufficient responsiveness of a 5-minute walk test and sufficient reliability of distance walked during the 6-minute walk test. Very low to low-level evidence limits further understanding of measurement properties for a wide range of physical measures.

**Table 4. Summary of findings—Reliability and measurement error.**

| Physical outcome measure / physical measure | Measurement instrument | Study | Reliability | | | | Measurement Error | | | |
|---|---|---|---|---|---|---|---|---|---|---|
| | | | Result (Rating) | Risk of bias | Overall rating [a] | Overall quality of evidence | Result (Rating) | Risk of bias | Overall rating [a] | Overall quality of evidence |
| **Impairment-based physical outcome measures** | | | | | | | | | | |
| *None studied* | | | | | | | | | | |
| **Performance-based physical outcome measures** | | | | | | | | | | |
| **Physical outcome measure: 5 repetitions sit to stand** | | | | | | | | | | |
| Physical measure: Time to complete | Stopwatch | Staartjes et al (2018) [60] | Test-retest reliability: ICC: 0.97 (95% CI 0.94–0.98), (+) | Inadequate | + | Low | SEM: 1.47 (?, MIC NR) | Inadequate | ? | Low |
| | Digital timer | Staartjes et al (2019) [61] | Inter-rater reliability: ICC$_{2,2}$: 0.996 (95% CI 0.993–0.998), (+) | Inadequate | | | 95% LoA: −0.81–0.51 (?, MIC NR) | Inadequate | | |
| | Clinic & Unsupervised tests: NR, Tele-supervised: Digital timer | Staartjes et al (2019) [61] | Test-rest reliability in clinic vs unsupervised: r: 0.94 (95% CI 0.91–0.96), (+) Test-retest reliability in clinic vs tele-supervised: r: 0.90 (95% CI 0.83–0.94), (+) | Inadequate | | | Not studied | | | |
| **Physical outcome measure: 6-minute walk test** | | | | | | | | | | |
| Physical measure: Distance walked | 6WT app | Zeitlberger et al (2022) [38] | Test-retest reliability: ICC: 0.82 (95% CI 0.75–0.87), (+) | Doubtful | + | Moderate | SEM: 58.3 (?, MIC NR) | Doubtful | ± | Low |
| | | Maldaner et al (2020) [55] | Test-retest reliability: ICC: 0.82 (95% CI 0.75–0.88), (+) | Doubtful | | | SEM: 58 (?, MIC NR) | Doubtful | | |
| | | Takenaka et al (2023) [67] | Not studied | | | | SEM: 34.5m at 6 months (+, MIC 100m) | Doubtful | | |
| Physical measure: Distance to first symptoms | 6WT app | Zeitlberger et al (2022) [38] | Test-retest reliability: ICC: 0.83 (95% CI 0.77–0.88), (+) | Doubtful | + | Very low | SEM: 85 (?, MIC NR) | Doubtful | ? | Very low |
| Physical measure: Time to first symptoms | 6WT app | Zeitlberger et al (2022) [38] | Test-retest reliability: ICC: 0.79 (95% CI 0.72–0.85), (+) | Doubtful | + | Very low | SEM: 59 (?, MIC NR) | Doubtful | ? | Very low |
| **Physical outcome measure: Treadmill test** | | | | | | | | | | |
| Physical measure: Time to first symptoms (pre-operative) | Treadmill, preferred speed and 1.2 mph, 0% incline | Deen et al (2000) [49] | Test-retest reliability: 1.2 mph: 0.90 Preferred speed: 0.98 (+) | Inadequate | + | Very low | Not studied | | | |

*(Continued)*

**Table 4.** (Continued)

| Physical outcome measure / physical measure | Measurement instrument | Study | Reliability | | | | Measurement Error | | | |
|---|---|---|---|---|---|---|---|---|---|---|
| | | | Result (Rating) | Risk of bias | Overall rating [a] | Overall quality of evidence | Result (Rating) | Risk of bias | Overall rating [a] | Overall quality of evidence |
| Physical measure: Time to first symptoms (post-operative) | Treadmill, preferred speed and 1.2 mph, 0% incline | Deen et al (2000) [49] | Test-retest reliability: 1.2 mph and preferred speed: NR* (?) | Inadequate | ? | Very low | Not studied | | | |
| Physical measure: Total ambulation time (pre-operative) | Treadmill, preferred speed and 1.2 mph, 0% incline | Deen et al (2000) [49] | Test-retest reliability: 1.2 mph: 0.89 Preferred speed: 0.96 (+) | Inadequate | + | Very low | Not studied | | | |
| Physical measure: Total ambulation time (post-operative) | Treadmill, preferred speed and 1.2 mph, 0% incline | Deen et al (2000) [49] | Test-retest reliability: 1.2 mph and preferred speed: NR* (?) | Inadequate | ? | Very low | Not studied | | | |
| **Activity in a natural environment physical outcome measures** | | | | | | | | | | |
| **Physical outcome measure: Step count** | | | | | | | | | | |
| Physical measure: Steps detected at thigh | ActivPAL3 Fitbit Flex Jawbone UP Move | Gilmore et al (2020) [73] | Not studied | | | | ActivPAL3: 23.2 Fitbit Flex: 35.8 Jawbone: 44.6 (?, MIC NR) | Doubtful | ? | Very low |
| Physical measure: Steps detected at wrist | Fitbit Flex Jawbone UP Move | Gilmore et al (2020) [73] | Not studied | | | | Fitbit Flex Total: 36.2 No gait aid: 43.3 Gait aid: 26.0 Jawbone: Total: 40.5 No gait aid: 58.1 Gait aid: Unable to calculate (no steps detected) (?, MIC NR) | Doubtful | ? | Very low |

[a] Rating according to COSMIN criteria for good measurement properties: (+) sufficient; (-) insufficient; (?) indeterminant; (±) inconsistent.

*Results not reported (NR) "most patients completed a full 15-minute examination, and there was little variability" (Deen et al., 2000). [49]

CI, Confidence interval; ICC, intraclass correlation coefficient; LoA, Limits of agreement; MIC, minimum important change; mph, miles per hour; NR, Not reported; SEM, Standard error of measurement.

## Stage one

The global importance of physical functioning in lumbar spinal surgery is emphasized by the breadth of countries represented in stage one and increasing number of publications across 40 years. This aligns with physical functioning advocated as a critical domain within core outcome sets for the past 25 years [8, 80, 81], and measurement instruments recommended for use within core outcome sets including well-established PROMs (ODI, Roland Morris Disability Questionnaire) [8, 80, 81]. However, stage one identified an extensive number of PROMs across all five categories of physical functioning [14] and physical

**Table 5. Summary of physical measures.**

| Physical outcome measure / physical measure | Measurement property | Overall Rating[a] | Overall quality of evidence |
|---|---|---|---|
| **Impairment-based physical outcome measures** | | | |
| **Physical outcome measure: Active range of movement** | | | |
| Physical measure: Lumbar extension | Responsiveness (Construct approach) [40] | ? | Very low |
| Physical measure: Lumbar, trunk and hip flexion and extension | Construct validity [44] | ? | Very low |
| | Responsiveness (Construct approach) [44] | ? | Very low |
| Physical measure: Lumbopelvic flexion and extension | Construct validity [45] | ? | Very low |
| Physical measure: Schober test | Responsiveness (Construct approach) [40] | ? | Very low |

Active range of movement was included in 2 studies [44, 45] (Inadequate RoB) evaluating construct validity of 2 physical measures. Computer assisted electronic inclinometer measures of lumbar spine, trunk and hip flexion and extension was compared to the Roland Morris Disability Questionnaire 1–2 days pre-operatively and 2 months post-operatively [44]. Dual bubble inclinometer measures of lumbopelvic flexion and extension was compared to the North American Spine Society Questionnaire (Disability and neurogenic symptom subscales) and a straight leg raise at Physical Therapy pre-operatively, first visit post-operatively and discharge [45]. Two studies [40, 44] (Inadequate RoB) evaluated responsiveness (construct approach) of 3 physical measures. Change in computer assisted electronic inclinometer measures of lumbar spine, trunk and hip flexion and extension was compared to change in Roland Morris Disability Questionnaire scores collected 1–2 days pre-operatively and 2 months post-operatively [44]. Change in Dualer goniometer measures of lumbar extension and the Schober test of lumbar flexion was compared to change in 15D health-related quality of life PROM collected 2 and 14 months post-operatively [40].

| | | | |
|---|---|---|---|
| **Physical outcome measure: Handgrip strength** | | | |
| Physical measure: Handgrip maximum voluntary contraction | Construct validity [41–43] | ? | Low |

Handgrip strength was included in 3 studies [41–43] (Inadequate RoB) evaluating construct validity. Maximum voluntary contraction of handgrip strength measured using a handheld dynamometer was compared to 5 PROMs, 8 physical measures and 13 radiological measures pre-operatively [41, 43] and one year post-operatively [42].

| | | | |
|---|---|---|---|
| **Physical outcome measure: Gait parameters** | | | |
| Physical measure: Two-step test | Construct validity [39] | ? | Low |
| | Criterion validity [39] | + | Low |
| Physical measure: Asymmetry of double support | Responsiveness (Construct approach) [31] | ? | Very low |
| Physical measure: Stride length | Responsiveness (Construct approach) [31] | ? | Very low |

The Two-step test was included in 1 study [39] evaluating construct validity (Inadequate RoB) and criterion validity (Doubtful RoB). For both construct and criterion validity, Two-step test results were compared to the TUG test 1 day pre-operatively. Asymmetry of double support and stride length were included in 1 study [31] (Inadequate RoB) evaluating responsiveness (construct approach). Change in RehabGait system physical measures of asymmetry of double support and stride length was compared to change in ODI collected 1 day pre-operatively and post-operatively (10 weeks, 12 months).

| | | | |
|---|---|---|---|
| **Performance-based physical outcome measures** | | | |
| **Physical outcome measure: 1-min stair climb** | | | |
| Physical measure: Number of stairs | Responsiveness (Construct approach) [53] | + | Moderate |

The 1-minute stair climb test was included in 1 study [53] (Very good RoB) evaluating responsiveness (construct approach). Change in the number of stairs climbed in 1 minute was compared to change in global perceived effect (construct-specific, general), physical measures (5-minute walk, 50-foot walk, TUG) and PROMs (ODI, back pain) collected 8–12 weeks pre-operatively and 6 months post-operatively.

| | | | |
|---|---|---|---|
| **Physical outcome measure: 5 repetitions sit to stand** | | | |
| Physical measure: Time to complete | Construct validity [54, 57, 60–62] | ? | Low |
| | Reliability (test retest and inter-rater) [60, 61] | + | Low |
| | Measurement error [60, 61] | ? | Low |

The 5 repetition sit to stand test was included in 5 studies [54, 57, 60–62] (Inadequate RoB) evaluating construct validity. Time to complete the test was compared to 8 PROMs pre-operatively. Two studies [60, 61] (Inadequate RoB) evaluated reliability. Test-retest reliability in a clinical environment, [60] inter-rater reliability of a tele-supervised test performed at home [61], and reliability between measures taken supervised in a clinic and unsupervised at home [61] were evaluated pre-operatively. Two studies [60, 61] (Inadequate RoB) evaluated measurement error. Time to complete the 5 repetition sit to stand test was compared using test-retest in a clinical environment [60] and inter-rater agreement of a tele-supervised test performed at home [61].

| | | | |
|---|---|---|---|
| **Physical outcome measure: 5-min walk test** | | | |
| Physical measure: Distance walked | Responsiveness (Construct approach) [53] | - | Moderate |

The 5-minute walk test was included in one study [53] (Very good RoB) evaluating responsiveness (construct approach). Change in the distance walked was compared to change in global perceived effect (construct-specific, general), physical measures (1-minute stair climb, 50-foot walk, TUG) and PROMs (ODI, back pain) collected 8–12 weeks pre-operatively and 6 months post-operatively.

| | | | |
|---|---|---|---|
| **Physical outcome measure: 6-minute walk test** | | | |

**Table 5.** (Continued)

| Physical outcome measure / physical measure | Measurement property | Overall Rating[a] | Overall quality of evidence |
|---|---|---|---|
| Physical measure: Distance walked | Construct validity [38, 55, 56, 70] | ? | Low |
| | Measurement error [38, 55, 67] | ± | Low |
| | Reliability (test retest) [38, 55] | + | Moderate |
| | Responsiveness (Criterion approach) [38, 56, 66, 67] | + | Low |
| | Responsiveness (Construct approach) [38, 56, 66, 67, 70] | ? | Low |
| | Responsiveness (Patient reported) [59] | ? | Very low |

Distance walked during the 6-minute walk test was included in 4 studies [38, 55, 56, 70] (Inadequate RoB) evaluating construct validity. Distance walked was measured using the 6WT app and compared to 7 PROMs and the Timed up and go test pre-operatively [38, 55, 56, 70], 6 weeks post-operatively [38, 56, 70], and 3 months post-operatively [70]. Three studies [38, 55, 67] (3 Doubtful RoB) evaluated measurement error. Distance walked measured using the 6WT app [38, 55] (Not reported [67]), was compared pre-operatively [55], pre-operatively to 6 weeks post-operatively [38] and pre-operatively to 6 months post-operatively [67]. Two studies [38, 55] (Doubtful RoB) evaluated reliability. Using the 6WT app, test-retest reliability was evaluated by comparing repeated measures pre-operatively [55] and pre-operatively to 6 weeks post-operatively [38]. Five studies [38, 56, 66, 67, 70] (Inadequate RoB) evaluated responsiveness (construct approach). Distance walked measured using the 6WT app [38, 56, 70] (Not reported [66, 67]) pre-operatively was compared to distance walked post-operatively at 6 weeks, [38, 56, 70] 3 months [70,] 6 months, [66, 67] and 12 months [67]. Four studies [38, 56, 66, 67] (3 Doubtful, 1 Inadequate RoB) evaluated responsiveness (criterion approach). Change in distanced walked measured using the 6WT app [38, 56] (Not reported [66, 67]) pre-operatively was compared to change in the Zurich Claudication Questionnaire satisfaction scale collected 6 weeks post-operative [38, 56] and the ODI collected post-operatively at 6 months [66, 67] and 12 months [67]. One study [59] (Inadequate RoB) evaluated patient-reported responsiveness. Following use of the 6WT app and completing 2 PROMs (Zurich claudication questionnaire, COMI) pre-operatively and 6 weeks post-operatively, patients completed the survey question "Which instrument do you consider best in detecting differences in your symptoms".

| Physical measure: Distance to first symptoms | Construct validity [38] | ? | Very low |
|---|---|---|---|
| | Measurement error [38] | ? | Very low |
| | Reliability (test retest) [38] | + | Very low |
| | Responsiveness (Criterion approach) [38] | + | Very low |
| | Responsiveness (Construct approach) [38] | ? | Very low |

Distance to first symptoms during the 6-minute walk test was included in 1 study [38] (Inadequate RoB) evaluating construct validity. Using the 6WT app, distance to first symptoms was compared to 5 PROMs pre-operatively and 6 weeks post-operatively. One study [38] (Doubtful RoB) evaluated reliability and measurement error. Using the 6WT app, test-retest reliability and measurement error were evaluated by comparing repeated measures pre-operatively to 6 weeks post-operatively. One study [38] (Inadequate RoB) evaluated responsiveness (criterion and construct approaches). For criterion approach, change in distance to first symptoms measured using the 6WT app was compared to change in the Zurich Claudication Questionnaire satisfaction scale collected pre-operatively and 6 weeks post-operatively. For construct approach, distance to first symptoms measured using the 6WT app pre-operatively was compared to distance to first symptoms 6 weeks post-operatively.

| Physical measure: Time to first symptoms | Construct validity [38] | ? | Very low |
|---|---|---|---|
| | Measurement error [38] | ? | Very low |
| | Reliability (test retest) [38] | + | Very low |
| | Responsiveness (Criterion approach) [38] | - | Very low |
| | Responsiveness (Construct approach) [38] | ? | Very low |

Time to first symptoms during the 6-minute walk test was included in 1 study [38] (Inadequate RoB) evaluating construct validity. Using the 6WT app, time to first symptoms was compared to 5 PROMs pre-operatively and 6 weeks post-operatively. One study [38] (Doubtful RoB) evaluated reliability and measurement error. Using the 6WT app, test-retest reliability and measurement error were evaluated by comparing repeated measures pre-operatively to 6 weeks post-operatively. One study [38] (Inadequate RoB) evaluated responsiveness (criterion and construct approaches). For criterion approach, change in time to first symptoms measured using the 6WT app was compared to change in the Zurich Claudication Questionnaire satisfaction scale collected pre-operatively and 6 weeks post-operatively. For construct approach, time to first symptoms measured using the 6WT app pre-operatively was compared to time to first symptoms 6 weeks post-operatively.

| **Physical outcome measure: 10-meter walk test** | | | |
|---|---|---|---|
| Physical measure: Walking speed | Construct validity [69] | ? | Very low |
| | Responsiveness (Construct approach) [69] | ? | Very low |

Walking speed during the 10-meter walk test was included in 1 study [69] (Inadequate RoB) evaluating construct validity. Walking speed was compared to the Pain Catastrophizing Scale pre-operatively and post-operatively at 3, 6 and 12 months. One study [69] (Inadequate RoB) evaluated responsiveness (Construct approach). Change in walking speed was compared to change in scores on the Pain Catastrophizing Scale collected pre-operatively between admission and surgery and postoperatively at 12 months.

| **Physical outcome measure: 50-foot walk test** | | | |
|---|---|---|---|
| Physical measure: Time to complete | Responsiveness (Construct approach) [53] | + | Moderate |

The 50-foot walk test was included in 1 study [53] (Very good RoB) evaluating responsiveness (construct approach). Change in the time to complete was compared to change in global perceived effect (construct-specific, general), physical measures (1-minute stair climb, 5-minute walk, TUG) and PROMs (ODI, back pain) collected 8–12 weeks pre-operatively and 6 months post-operatively.

*(Continued)*

**Table 5.** (Continued)

| Physical outcome measure / physical measure | Measurement property | Overall Rating[a] | Overall quality of evidence |
|---|---|---|---|
| **Physical outcome measure: Modified Sorensen test** | | | |
| Physical measure: Time to exhaustion | Construct validity [47, 48] | ? | Very low |

The modified Sorensen test was included in 2 studies [47, 48] (Inadequate RoB) evaluating construct validity. Time to exhaustion was compared to 14 PROMs and electromyography measures (L1 and L5 slope) collected 2 weeks to 1 day pre-operatively and post-operatively (4 weeks,[47] 2 years [48]).

| | | | |
|---|---|---|---|
| **Physical outcome measure: Self-paced walking test** | | | |
| Physical measure: Distance walked | Responsiveness (Construct approach) [68] | ? | Very low |
| Physical measure: Time walked | Responsiveness (Construct approach) [68] | ? | Very low |

The self-paced walking test was included in 1 study [68] (Inadequate RoB) evaluating responsiveness (construct approach) of distance and time walked. Change in distance walked and time walked during the test was compared to change in 6 PROMs collected 1 week pre-operatively and 6 weeks post-operatively.

| | | | |
|---|---|---|---|
| **Physical outcome measure: Timed up and go** | | | |
| Physical measure: Time to complete | Construct validity [46, 50, 56, 57, 63–65] | ? | Very low |
| | Responsiveness (Construct approach) [51, 53, 56] | ± | Low |
| | Responsiveness (Criterion approach) [56] | - | Very low |

The TUG test was included in 7 studies [46, 50, 56, 57, 63–65] (6 Inadequate, 1 Doubtful RoB) evaluating construct validity. Using the TUG app[56,63,65] or stopwatch [64] (Not reported [46, 50, 57]), time to complete the TUG was compared to 17 PROMs, radiological measures (Modic and Pfirrman classification), clinician-reported measures (Charlson Comorbidity Index, American Society of Anaesthesiology grading) and the 6-minute walk test pre-operatively [46, 50, 56, 57, 63–65] and 6 weeks post-operatively [56]. Three studies [51, 53, 56] (2 Inadequate, 1 Very good RoB) evaluated responsiveness (construct approach) and one study [56] (Doubtful RoB) evaluated responsiveness (criterion approach). For construct approach, change in time to complete the TUG was compared to change in 7 PROMs, global perceived effect (construct-specific, generic) and physical measures (1-minute stair climb, 5-minute walk, 50-foot walk) collected pre-operatively and post-operatively (3 days, [51] 6 weeks, [51, 56] 6 months [53]). Time to complete the TUG pre-operatively was also compared to time to complete 6 weeks post-operatively [56]. For criterion approach,[56] change in time to complete the TUG was compared to change in the Zurich Claudication Questionnaire satisfaction scale collected pre-operatively and 6 weeks post-operatively.

| | | | |
|---|---|---|---|
| **Physical outcome measure: Treadmill test** | | | |
| Physical measure: Maximum walking distance | Construct validity [52, 58] | ? | Low |
| Physical measure: Maximum walking time | Construct validity [58] | ? | Very low |
| | Reliability (test-retest, Pre-operative) [49] | + | Very low |
| | Reliability (Post-operative) [49] | ? | Very low |
| Physical measure: Time to first symptoms | Construct validity [58] | ? | Very low |
| | Reliability (test-retest, Pre-operative) [49] | + | Very low |
| | Reliability (test-retest, Post-operative) [49] | ? | Very low |
| Physical measure: Distance to first symptoms | Construct validity [58] | ? | Very low |

A treadmill test was included in 2 studies [52, 58] (Inadequate RoB) evaluating construct validity of 4 physical measures. Treadmill test maximum walking distance was compared to 6 PROMs, radiological measures (degenerative findings, minimum area of dural sac, thecal sac cross-sectional area), and treadmill test time to first symptoms pre-operatively [58] and post-operatively at 6 months [58] and 11 years [52]. Treadmill test maximum walking time, time to first symptoms and distance to first symptoms were compared to thecal sac cross-sectional area pre-operatively and 6 months post-operatively [58]. One study [49] (Inadequate RoB) evaluated reliability. Test-retest reliability of treadmill test maximum walking time and time to first symptoms was evaluated pre-operatively and 6 months post-operatively.

| | | | |
|---|---|---|---|
| **Physical outcome measure: Trunk muscle endurance** | | | |
| Physical measure: Repetitive arch-ups until exhaustion | Responsiveness (Construct approach) [40] | ? | Very low |
| Physical measure: Repetitive sit-ups until exhaustion | Responsiveness (Construct approach) [40] | ? | Very low |

Trunk muscle endurance was included in 1 study [40] (Inadequate RoB) evaluating responsiveness (construct approach) of 2 physical measures. Change in the number of repetitive arch ups and sit ups until exhaustion was compared to change in 15D health-related quality of life PROM collected 2 and 14 months post-operatively.

| | | | |
|---|---|---|---|
| **Activity in a natural environment physical outcome measures** | | | |
| **Physical outcome measure: Step counts** | | | |
| Physical measure: Steps per day | Construct validity [71, 74, 78] | ? | Very low |
| | Responsiveness (Construct approach) [75] | ? | Very low |

Steps per day was included in 3 studies [71, 74, 78] (Inadequate RoB) evaluating construct validity using a Fitbit Flex 2, [71] Fitbit Charge [74] and Mi Band [78]. The number of steps taken per day was compared to 6 PROMs collected pre-operatively and a range of post-operative time points (1–7 days [74], 2 weeks [71], 1 month [71], 3 months [71, 78], 6 months [71], 12 months [78]). One study [75] (Inadequate RoB) evaluated responsiveness (construct approach) using a Fitbit Zip. Change in the number of steps per day was compared to change in 5 PROMs collected 7 days pre-operatively and post-operatively at 1, 2 and 3 months.

| | | | |
|---|---|---|---|
| Physical measure: Steps detected at thigh | Criterion validity [73] | ± | Very low |
| | Measurement error [73] | ? | Very low |

*(Continued)*

**Table 5.** (Continued)

| Physical outcome measure / physical measure | Measurement property | Overall Rating[a] | Overall quality of evidence |
|---|---|---|---|
| Physical measure: Steps detected at wrist | Criterion validity [73] | - | Very low |
| | Measurement error [73] | ? | Very low |

Steps detected at the thigh and wrist was included in 1 study [73] (Doubtful RoB) evaluating criterion validity using ActivPAL3 (thigh only), Fitbit Flex and Jawbone UP Move. Steps detected by the activity monitors was compared to observed step count on the second or third day post-operatively. One study [73] (Doubtful RoB) evaluated measurement error using ActivPAL3 (thigh only), Fitbit Flex and Jawbone UP Move. Steps detected were compared to observed step count on the second or third day post-operatively.

| **Physical outcome measure: Gait Posture Index** | | | |
|---|---|---|---|
| Physical measure: Gait Posture Index | Responsiveness (Construct approach) [72, 76] | ? | Very low |

Gait Posture Index was included in 2 studies [72, 76] (Inadequate RoB) evaluating responsiveness (construct approach) using participants personal devices (e.g., Apple watch, Garmin) or Mi Band 2. Change in Gait Posture Index was compared to change in ODI and patient satisfaction pre-operatively and 3 months post-operatively.

| **Physical outcome measure: Distance walked per day** | | | |
|---|---|---|---|
| Physical measure: Distance walked per day | Construct validity [79] | ? | Very low |
| | Responsiveness (Construct approach) [75] | ? | Very low |

Distance walked per day (mile / day) was included in 1 study [79] (Inadequate RoB) evaluating construct validity using Apple Health activity data from an Apple iOS personal smartphone. Distanced walked per day was compared to distance walked during the 6-minute walk test and 3 PROMs pre-operatively and post-operatively at 6 and 12 weeks. One study [75] (Inadequate RoB) evaluated responsiveness (construct approach) of distance walked per day (km / day) using a Fitbit Zip. Change in distance per day was compared to change in 5 PROMs 7 days pre-operatively and post-operatively at 1, 2 and 3 months.

| **Physical outcome measure: Gait cycles** | | | |
|---|---|---|---|
| Physical measure: Gait cycles per day | Construct validity [77] | ? | Very low |
| Physical measure: Gait cycles per hour | Construct validity [77] | ? | Very low |
| Physical measure: Gait intensities per day | Construct validity [77] | ? | Very low |

Number of gait cycles per day, gait cycles per hour, and gait intensities (>40 gait cycles per minute) were included in 1 study [77] (Inadequate RoB) evaluating construct validity using the StepWatch 3. Number of gait cycles were compared to 4 PROMs and 4 radiological measures pre-operatively and post-operatively at 3 and 12 months.

[a] Rating according to COSMIN criteria for good measurement properties: (+) sufficient; (-) insufficient; (?) indeterminant; (±) inconsistent.

PROMs, Patient reported outcome measures; ROB, Risk of bias; ODI; Oswestry Disability Index; TUG, Timed up and go.

measures across 15 level-two ICF categories [15]. Use of a range of measures aligns with previous systematic reviews highlighting limited and inconsistent implementation of recommendations for standardizing outcome measures in LBP clinical trials [82], and substantiates physical functioning as a multidimensional construct not best measured with a single PROM. Support for PROMs and physical measures to evaluate physical functioning in other musculoskeletal disease is strong, including international recommendations within clinical trials [17, 19, 83]. In lumbar spinal surgery, there is emerging evidence demonstrating the value of physical measures (important to patients [59], responsive to change [38, 53, 56, 59, 66], predictive of outcomes [84]). However, recommendations for their use in LBP populations do not exist.

Establishing consensus on appropriate physical measures of physical functioning is required to enable comparisons of interventions and outcomes. Stage one results highlight an illustrative example. Walking was frequently evaluated with PROMs and physical measures, aligning with previous research emphasizing walking as an important component of rehabilitation in lumbar spinal surgery [85–88]. However, with 23 different stand-alone PROM questions/response options to measure walking capacity and 21 different walking physical measures, it is nearly impossible to compare across studies. Action is required to standardize measurement of physical functioning outcomes.

## Stage two

Sufficient measurement properties are required for recommending measures to use in research and clinical practice [89, 90]. While 134 physical measures were identified in stage one, only 34 had investigations of measurement properties. Few recommendations can be made because of indeterminant measurement properties and mostly very low to low-level evidence. The strongest evidence supports performance-based measures, with moderate-level evidence for sufficient responsiveness of 1-minute stair climb and 50-foot walk tests, insufficient responsiveness of the 5-minute walk test and sufficient reliability of distance walked during the 6-minute walk test. Very low to low-level evidence limits further understanding of measurement properties. However, results illustrate promise for a range of physical measures, for example when hypotheses are appropriately defined for construct approaches to validity and responsiveness [53, 65] or no hypothesis is required to evaluate the measurement property [38, 39, 49,55, 56, 60, 61, 66, 67, 73]. according to COSMIN [20]. While emerging evidence supports the value of physical measures, at present few clear recommendations can be made. Prospective, low risk of bias studies are required.

A challenge in conducting this systematic review was inconsistent terminology and poor reporting in included studies. Inconsistent terminology led to lack of clarity in the measurement property under investigation. For example, three studies [38, 55, 65] reported an investigation of content validity, however study designs and statistical methodologies aligned with the COSMIN definition of hypothesis testing for construct validity. Poor reporting strongly contributed to indeterminant measurement properties, high RoB and very low to low-level evidence. It precluded COSMIN recommendations to derive hypotheses for study authors in the absence of a priori hypotheses in hypothesis testing approaches because of insufficient information about the expected direction and strength of associations. COSMIN guidelines were designed to enable systematic reviews but can also be used to inform terminology and reporting of studies on measurement properties. However, this is not common, as inconsistent terminology and poor reporting are challenges identified in other systematic reviews investigating measurement properties of physical measures [91, 92]. COSMIN has published reporting guidelines for studies on measurement properties [93], however lessons learned from Consolidated Standards of Reporting Trials (CONSORT) suggest at least 15 years may be required for widespread implementation [94]. PRISMA-COSMIN reporting guidelines for systematic reviews of measurement properties[95] will hopefully hasten the process to draw clear conclusions about measurement properties of physical measures.

This systematic review demonstrates expanding use of digital technologies in remote monitoring of physical functioning. Several digital technologies were used (personal devices [38, 55, 56, 59, 61, 65, 72, 76], low-cost consumer-grade [71–78] and research-grade wearables [31, 73]) to measure impairments, performance and activity in a natural environment, with unique combinations in some studies. For example, free smartphone applications enabled digital self-assessments of performance-based measures (e.g., 6-minute walk) in a natural environment, but with limited and very low to low-level evidence supporting measurement properties [38, 56, 59, 65]. However, one unique study of patient-reported responsiveness indicated that patients perceive a smartphone application that measured 6-minute walk performance was better at detecting changes in their symptoms compared to PROMs (very low-level evidence) [59]. In selecting appropriate digital technologies in lumbar spinal surgery, important factors to consider are wear position and gait aids, as both influence validity of step count detection (very low-level evidence) [73]. Digital technologies are promising solutions to enable personalized remote monitoring, aid clinical reasoning and inform tailored interventions [96], however their measurement properties are largely unknown, necessitating low RoB studies.

## Strengths and limitations

This robust systematic review used a rigorous two-staged approach to identify physical functioning outcome measures, enabling a comprehensive search for measurement properties of physical measures of physical functioning. It is limited by heterogeneity in physical measures, indeterminant measurement properties and RoB across included studies. Poor reporting of studies was a key issue. Important findings may have been missed with the exclusion of non-English full texts (108 in stage one, 1 in stage two). Limitations prevent clear recommendations for a range of physical measures of physical functioning in the lumbar spinal surgery population.

## Conclusions

While many physical measures are used to evaluate physical functioning in the lumbar spinal surgery population, few have investigations of measurement properties. Research to date is overall low quality, consisting of high RoB studies with inconsistent use of terminology and poor reporting. The strongest evidence supports performance-based measures, with moderate-level evidence for sufficient responsiveness of the 1-minute stair climb and 50-foot walk tests, insufficient responsiveness of the 5-minute walk test and sufficient reliability of distance walked during the 6-minute walk test. There is promise for physical measures of physical functioning to demonstrate sufficient measurement properties, but few clear recommendations can be made. Knowledge of measurement properties is essential in establishing consensus on appropriate physical measures of physical functioning to evaluate effectiveness of interventions for lumbar spinal surgery populations and inform clinical practice. Prospective low RoB studies are required owing to emerging evidence demonstrating the value of physical measures.

## Supporting information

**S1 Appendix. Search strategies.**
(DOCX)

**S2 Appendix. Rating criteria for measurement properties.**
(DOCX)

**S3 Appendix. Articles excluded at full text stage (stage 2).**
(DOCX)

**S4 Appendix. Stage one results.**
(XLSX)

**S5 Appendix. Summary of stage one results.**
(DOCX)

**S6 Appendix. Stage two results.**
(DOCX)

**S1 Checklist. PRISMA checklist.**
(DOCX)

## Acknowledgments

Study authors thank Meagan Stanley for peer-reviewing the stage one electronic search strategy and Keerthana Jayaprakash for administrative support in stage one.

## Patient and public involvement

The spinal pain research Patient Partner Advisory Group (PPAG) in the School of Physical Therapy at Western University has shaped our program of research methodology focused on physical measures of physical functioning. Systematic review results have been discussed with the PPAG to inform development of future research projects.

## Author Contributions

**Conceptualization:** Katie L. Kowalski, Michael J. Lukacs, Alison Rushton.

**Data curation:** Katie L. Kowalski.

**Formal analysis:** Katie L. Kowalski, Jai Mistry.

**Investigation:** Katie L. Kowalski, Jai Mistry, Anthony Beilin, Maren Goodman.

**Methodology:** Katie L. Kowalski, Jai Mistry, Maren Goodman, Alison Rushton.

**Project administration:** Katie L. Kowalski.

**Resources:** Maren Goodman, Alison Rushton.

**Supervision:** Alison Rushton.

**Validation:** Katie L. Kowalski.

**Visualization:** Katie L. Kowalski.

**Writing – original draft:** Katie L. Kowalski.

**Writing – review & editing:** Katie L. Kowalski, Jai Mistry, Anthony Beilin, Maren Goodman, Michael J. Lukacs, Alison Rushton.

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
