## [Decision Letter · Decision Letter 0]

27 May 2024

PONE-D-24-11136Physical functioning in the lumbar spinal surgery population: a systematic review and narrative synthesis of outcome measures and measurement properties of the physical measuresPLOS ONE

Dear Dr. Kowalski,

Thank you for submitting your manuscript to PLOS ONE. After careful consideration, we feel that it has merit but does not fully meet PLOS ONE’s publication criteria as it currently stands. Therefore, we invite you to submit a revised version of the manuscript that addresses the points raised during the review process.

**ACADEMIC EDITOR: **

**There are some comments raised from the reviewers, you will find these comments in the reviewers' comments section. I would like to invite the authors to respond to these comments one by one in details before submitting the revised version of the manuscript.**

We look forward to receiving your revised manuscript.

Kind regards,

Emad A. Aboelnasr, Ph.D

Academic Editor

PLOS ONE

Reviewers' comments:

Reviewer's Responses to Questions

**Comments to the Author**

1. Is the manuscript technically sound, and do the data support the conclusions?

Reviewer #1: Yes

Reviewer #2: Yes

2. Has the statistical analysis been performed appropriately and rigorously? 

Reviewer #1: N/A

Reviewer #2: N/A

3. Have the authors made all data underlying the findings in their manuscript fully available?

Reviewer #1: Yes

Reviewer #2: Yes

4. Is the manuscript presented in an intelligible fashion and written in standard English?

Reviewer #1: Yes

Reviewer #2: Yes

5. Review Comments to the Author

Reviewer #1: The review is relevant and professionally written. To avoid confusing study design with current study design, it is advised to change the sub-heading "study design" to "design of included study". Page 7, line 127.

Reviewer #2: This Review was carried out very seriously and conscientiously. Nevertheless, there are a few points for improvement.

figure 1: is not readable, please replace it with a high-resolution figure.

Page 4 / Line 54/55: First line therapy for LBP is conservative treatment, Please state why you concentrated your review on surgigal intervention. Does outcome measures differ between conservativ and surgical interventions?

objectives:

- the first objective includes patient reported and physical outcome measures. The sencond objective only physical outcomes. Why did you include PROMs in the first step but not in the second step?

I- In addition to the measurement properities, you also assessed interpretability and feasibility. Pleas add this part to your objectives,

Page 9 / 184/185: Why are studies using a criterion approach considered responsivenss. This is not in accordance with COSMIN. Pleas refer to the definition of COSMIN (for validity and responsiveness)

Page 10 / Line 190: what is meant by "study authors" (authors of this review or of the included studies)? Why would formulating hypothesis elevate ROB? Pleas explaain this connection.

Page 14 / Line261-264: What is the difference between physical measures and physical outcone measures? Pleas define these two terms.

Table 1: please indicate what sort of reliability was evaluated (intra, inter, test-retest)

Table 4: dito table 1

6. PLOS authors have the option to publish the peer review history of their article (what does this mean?). If published, this will include your full peer review and any attached files.

Reviewer #1: No

Reviewer #2: No

---

## [Author Response · Author response to Decision Letter 0]

3 Jun 2024

We have incorporated feedback into the manuscript from the two reviewers, which has improved its quality, clarity, and conciseness. We have addressed each feedback point from the two reviewers in the submitted document titled “Kowalski et al Point-by-point response to reviewers”. The corresponding changes to the manuscript and associated tables / figures (where relevant) are highlighted using tracked changes, and uploaded with a file naming convention indicating as such. 

Point by point responses: 

Reviewer 1 comment

The review is relevant and professionally written. To avoid confusing study design with current study design, it is advised to change the sub-heading "study design" to "design of included study". Page 7, line 127. 

Author’s Response

Thank you for this feedback. 

We have clarified this eligibility criteria sub-heading to reflect study design of included studies.

Reviewer 2 comment

figure 1: is not readable, please replace it with a high-resolution figure. 

Author’s Response

A new Fig 1 file has been uploaded with the following file size properties: 5760x7811 pixels, 768 dpi

Reviewer 2 comment

Page 4 / Line 54/55: First line therapy for LBP is conservative treatment, Please state why you concentrated your review on surgigal intervention. Does outcome measures differ between conservativ and surgical interventions? 

Author’s Response

Thank you for this suggestion. We have added a statement starting on line 55 highlighting how population-specific outcome measures are recommended for use when measuring treatment outcomes for specific clinical populations and when focusing on the individual, an important component of providing patient-centered care. 

Reviewer 2 comment

objectives:

- the first objective includes patient reported and physical outcome measures. The sencond objective only physical outcomes. Why did you include PROMs in the first step but not in the second step? 

Author’s Response

Thank you for pointing this out. There are 2 reasons for this decision. 1: Systematic reviews of PROM measurement properties exist, and we have added this to the end of the intro (Line 93), yet there is still no contemporary comprehensive resource outlining all PROMs of physical functioning. 2: This is also a pragmatic decision, as the systematic review would have been very large had evaluations of PROMs and physical measures been included in objective 2. 

Reviewer 2 comment

I- In addition to the measurement properities, you also assessed interpretability and feasibility. Pleas add this part to your objectives, 

Author’s Response

We have added to objective 2 that interpretability and feasibility were described (Line 100 and 27-28 in abstract). We have also added to the introduction that COSMIN recommends considering interpretability and feasibility when selecting outcome measures, as further support to describing interpretability and feasibility as one step in the process of conducting a systematic review of measurement properties (Line 89).

Reviewer 2 comment

Page 9 / 184/185: Why are studies using a criterion approach considered responsivenss. This is not in accordance with COSMIN. Pleas refer to the definition of COSMIN (for validity and responsiveness) 

Author’s Response

We used the terminology of criterion approach for responsiveness as this is how it is described in the COSMIN risk of bias checklist for responsiveness (Box 10a), for example in studies where area under the receiver operating characteristic curve was used to evaluate responsiveness. This terminology is also useful to differentiate this approach from hypothesis testing construct approaches for responsiveness. We have added the reference for the COSMIN risk of bias checklist to support use of this terminology. 

Reviewer 2 comment

Page 10 / Line 190: what is meant by "study authors" (authors of this review or of the included studies)? Why would formulating hypothesis elevate ROB? Pleas explaain this connection. 

Author’s Response

We have clarified ‘study authors’ as ‘authors of included studies’ and that lack of a priori hypotheses introduces threats to internal validity of included studies and therefore this systematic review would have provided an inaccurate representation of the quality of the literature (Lines 196-199).

Reviewer 2 comment

Page 14 / Line261-264: What is the difference between physical measures and physical outcone measures? Pleas define these two terms. 

Author’s Response

We have added definitions for physical measures and physical outcome measures to supplement the examples already provided (Lines 268-270).

Reviewer 2 comment

Table 1: please indicate what sort of reliability was evaluated (intra, inter, test-retest); Table 4: dito table 1 

Author’s Response

In Tables 1 and 4, we have further described the type of reliability being evaluated.

---

## [Decision Letter · Decision Letter 1]

26 Jun 2024

Physical functioning in the lumbar spinal surgery population: a systematic review and narrative synthesis of outcome measures and measurement properties of the physical measures

PONE-D-24-11136R1

Dear Dr.,Katie

We’re pleased to inform you that your manuscript has been judged scientifically suitable for publication and will be formally accepted for publication once it meets all outstanding technical requirements.

Kind regards,

Emad A. Aboelnasr, Ph.D

Academic Editor

PLOS ONE

Additional Editor Comments (optional):

Reviewers' comments:

Reviewer's Responses to Questions

**Comments to the Author**

1. If the authors have adequately addressed your comments raised in a previous round of review and you feel that this manuscript is now acceptable for publication, you may indicate that here to bypass the “Comments to the Author” section, enter your conflict of interest statement in the “Confidential to Editor” section, and submit your "Accept" recommendation.

Reviewer #1: All comments have been addressed

Reviewer #2: All comments have been addressed

2. Is the manuscript technically sound, and do the data support the conclusions?

Reviewer #1: (No Response)

Reviewer #2: Yes

3. Has the statistical analysis been performed appropriately and rigorously? 

Reviewer #1: (No Response)

Reviewer #2: Yes

4. Have the authors made all data underlying the findings in their manuscript fully available?

Reviewer #1: (No Response)

Reviewer #2: Yes

5. Is the manuscript presented in an intelligible fashion and written in standard English?

Reviewer #1: (No Response)

Reviewer #2: Yes

6. Review Comments to the Author

Reviewer #1: (No Response)

Reviewer #2: no more comments, all comments in from the previous round of review have been addressed satisfactorily.

7. PLOS authors have the option to publish the peer review history of their article (what does this mean?). If published, this will include your full peer review and any attached files.

Reviewer #1: No

Reviewer #2: No

---

## [Editor Report · Acceptance letter]

28 Jun 2024

PONE-D-24-11136R1 

PLOS ONE

Dear Dr. Kowalski, 

I'm pleased to inform you that your manuscript has been deemed suitable for publication in PLOS ONE. Congratulations! Your manuscript is now being handed over to our production team.

Kind regards, 

on behalf of

Dr. Emad A. Aboelnasr 

Academic Editor

PLOS ONE